# DisDet: Exploring Detectability of Backdoor Attack on Diffusion Models

**Yang Sui**[1]   **Huy Phan**[1]   **Jinqi Xiao**[1]   **Tianfang Zhang**[1]   **Zijie Tang**[2]
**Cong Shi**[3]   **Yan Wang**[2]   **Yingying Chen**[1]   **Bo Yuan**[1]
[1]*Rutgers University*   [2]*Temple University*   [3]*New Jersey Institute of Technology*

*ys764@scarletmail.rutgers.edu*

**Reviewed on OpenReview:** *https://openreview.net/forum?id=SfqCaAOF1S*

## Abstract

In the exciting generative AI era, the diffusion model has emerged as a very powerful and widely adopted content-generation tool. Very recently, some pioneering works have shown the vulnerability of the diffusion model against backdoor attacks, calling for in-depth analysis and investigation of the security challenges. In this paper, we explore the detectability of the poisoned noise input for the backdoored diffusion models, an important performance metric yet little explored in the existing works. Starting from the perspective of a defender, we first analyze the distribution discrepancy of the trigger pattern in the existing diffusion backdoor attacks. Based on this finding, we propose a trigger detection mechanism that can effectively identify the poisoned input noise. Then, from the attack side, we propose a backdoor attack strategy that can learn the *unnoticeable trigger* to evade our proposed detection scheme. Our empirical evaluations across various diffusion models and datasets demonstrate the effectiveness of the proposed trigger detection and detection-evading attack strategy. For trigger detection, our distribution discrepancy-based solution can achieve a 100% detection rate for the Trojan triggers used in the existing works. For evading trigger detection, our proposed stealthy trigger design approach performs end-to-end learning to make the distribution of poisoned noise input approach that of benign noise, enabling nearly 100% detection pass rate with very high attack and benign performance for the backdoored diffusion models.

## 1 Introduction

Recently, the diffusion model has emerged as a prevalent generative AI technique for content creation and editing across various data modalities, including image, video, speech, text, etc. Built on the core principle originating from non-equilibrium thermodynamics, a diffusion model aims to learn to generate the target probability distribution via constructing and reverting a series of latent variables. Thanks to its solid theoretical foundations and training stability, to date the diffusion models have been widely used in various generative tasks, such as image generation Ho et al. (2020); Song et al. (2020a); Bordes et al. (2022); Chao et al. (2021); Karras et al. (2022), text-to-image synthesis Rombach et al. (2022); Saharia et al. (2022a); Ramesh et al. (2022); Singh et al. (2023), image editing Meng et al. (2021); Couairon et al. (2022); Brooks et al. (2023), image inpainting Lugmayr et al. (2022), super-resolution Rombach et al. (2022); Saharia et al. (2022b); Choi et al. (2021) and video generation Ho et al. (2022a); Esser et al. (2023); Harvey et al. (2022).

Since diffusion models have already served as the backbone components in many real-world applications, the corresponding security issues have become a potentially challenging risk that requires special attention. In particular, the vulnerabilities of diffusion models under *backdoor attack*, as a common and essential attack strategy against the existing classification models Chen et al. (2017); Gu et al. (2019); Nguyen & Tran

(2020b); Liu et al. (2020); Doan et al. (2021); Phan et al. (2022a); Zheng et al. (2023); Yuan et al. (2023), should be carefully reviewed and studied in the emerging generative AI era.

Despite the current research prosperity of the applications of diffusion models, the security challenges of this vital technique in the backdoor attack scenario are still under-explored. To date, only very few works investigate the backdoor attack tailored to diffusion models. In particular, Chen et al. (2023); Chou et al. (2023), as the representative works in this topic, propose the forward and backward processes of the backdoored diffusion models, demonstrating that the currently representative diffusion models can be attacked to generate the images in a target category or even targeted fixed images, with the presence of poisoned input noise.

Although these existing works reveal the feasibility of implanting Trojans into the diffusion models, we argue that the vulnerability and robustness of diffusion models under backdoor attack are still under-explored. In particular, the prior works mainly craft the poisoned noise input, leaving a blank in exploring the *detectability* of the embedded Trojans. Such exploration of the stealthiness of the Trojan trigger is very critical in both attacker and defender aspects.

In this paper, we explore the detectability of poisoned noise input for the backdoored diffusion models, from both attacker and defender perspectives. We first analyze the existing fixed trigger pattern, discovering the distribution discrepancy of noise input to the Gaussian noise. Based on this finding, we develop a trigger detection mechanism that can effectively identify the poisoned input noise. We then take a further step to propose a backdoor attack strategy that can learn the stealthy trigger of the proposed detection scheme. Overall, the contributions of this paper are summarized as follows:

- We explore the detectability of trigger patterns in the state-of-the-art diffusion model backdoor attacks. By analyzing the distribution discrepancy of the noise input, we propose a distribution-based detection mechanism that can identify the poisoned noise input of the backdoored diffusion models.

- We then develop a backdoor attack strategy that can evade our proposed detection method. By performing end-to-end learning of the trigger pattern towards minimizing the distribution discrepancy, the poisoned noise input can exhibit a very similar distribution to the benign input, making the backdoor attack unnoticeable. We also optimize the training process of the stealthy trigger pattern to improve the benign and attack performance of the backdoored diffusion models.

- We perform empirical evaluations for different diffusion models across different datasets and demonstrate the effectiveness of the proposed trigger detection and detection-evading attack strategy. On the defender side, our proposed distribution-based detection method can achieve a 100% detection rate for the trigger patterns used in the existing works. On the attacker side, our proposed detection-evading trigger can enable nearly 100% detection pass rate and bring high attack and benign performance for the backdoored diffusion models.

*Notice that the threat model in the previous backdoored diffusion model Chen et al. (2023); Chou et al. (2023) assumes that 1) the attacker can control the training process of the diffusion model. 2) The users/attackers can have access to the input Gaussian noise of the diffusion model. We follow the same threat model to conduct our analysis in this paper.*

## 2 Related Works

**Diffusion Models.** Diffusion models have emerged as a powerful generative AI technique very recently. Compared with other deep generative models, diffusion models exhibit good training stability and better quality and diversity of the generated data, making them popularly adopted in a variety of generative tasks, e.g., image generation Ho et al. (2020); Song et al. (2020a); Ho et al. (2022b); Dhariwal & Nichol (2021); Liu et al. (2023); Bordes et al. (2022); Chao et al. (2021); Karras et al. (2022), video generation Ho et al. (2022a); Esser et al. (2023); Harvey et al. (2022), text-to-image synthesis Rombach et al. (2022); Saharia et al. (2022a); Ramesh et al. (2022); Singh et al. (2023); Kumari et al. (2023); Gu et al. (2022); Zhang et al. (2023); Ruiz et al. (2023); Zhang et al. (2023) and fast sampling Song et al. (2020a); Salimans & Ho (2021); Sui et al.

(2024); Wu et al. (2024); Lu et al. (2022). Diffusion models can be formulated in different ways, such as denoising diffusion probabilistic model (DDPM) Ho et al. (2020) and its variant DDIM Song et al. (2020a), noise conditional score network (NCSN) Song et al. (2020b) and latent diffusion model (LDM) Rombach et al. (2022). This paper focuses on the backdoor attack on DDPM/DDIM, as the most representative and fundamental diffusion model type.

**Backdoor Attacks on AI Models.** The research on launching backdoor attacks against AI models, especially the classification models, has been widely reported in the literature Chen et al. (2017); Gu et al. (2019); Nguyen & Tran (2020b); Liu et al. (2020); Doan et al. (2021); Saha et al. (2020); Li et al. (2021); Phan et al. (2022b); Salem et al. (2022); Nguyen & Tran (2020a). In this attack scenario, the adversary first poisons the training data to inject the backdoor into the model in the training phase. Then in the inference phase, the backdoored model behaves normally with the presence of benign input; while it will exhibit malicious behavior (e.g., misclassification) when the input is embedded with a Trojan trigger. Considering its natural stealthiness and severe damage, a series of backdoor defense approaches have been proposed Gao et al. (2019); Wang et al. (2019); Liu et al. (2018); Chen et al. (2019a;b); Tran et al. (2018); Li et al. (2020); Phan et al. (2024); Doan et al. (2023).

**Backdoor Attacks on Diffusion Models.** Unlike the extensive research on classification models, the backdoor attack for diffusion models is little explored yet. To date, the most two representative works are Chen et al. (2023); Chou et al. (2023), which for the first time demonstrate the feasibility of launching backdoor attack against the generative models. By adding a pre-defined trigger into the benign Gaussian noise input, the manipulated poisoned noise can prompt the backdoored diffusion model to generate a target image Chou et al. (2023); Chen et al. (2023) (e.g., Hello Kitty) or images belonging to a certain class Chen et al. (2023) (e.g., "horse") as desired by attackers. Because the adversary can leverage such malicious behavior to generate potentially offensive or illegal images, the vulnerability of diffusion models against backdoor attacks poses severe security challenges and risks.

## 3 Background

### 3.1 Diffusion Model

Diffusion model Sohl-Dickstein et al. (2015); Ho et al. (2020) is a type of deep generative model aiming to generate semantic-rich data from Gaussian noise. To realize such mapping, a diffusion model typically consists of forward *diffusion process* and backward *generative process*. Take the representative denoising diffusion probabilistic model (DDPM) Ho et al. (2020) as an example. In the diffusion process, an image $\mathbf{x}_0$ sampled from real data distribution $q(\mathbf{x}_0)$ is gradually diffused with the added random Gaussian noise over $T$ time steps. More specifically, this procedure generates a sequence of random variables $\mathbf{x}_1, \mathbf{x}_2, \cdots, \mathbf{x}_T$ in a Markov chain as $\mathbf{x}_t = \sqrt{1 - \beta_t}\mathbf{x}_{t-1} + \beta_t\boldsymbol{\epsilon}$ and $q(\mathbf{x}_t|\mathbf{x}_{t-1}) := \mathcal{N}(\mathbf{x}_t; \sqrt{1 - \beta_t}\mathbf{x}_{t-1}, \beta_t\mathbf{I})$, where $\beta_t$ is the pre-defined variance schedule and $\boldsymbol{\epsilon} \sim \mathcal{N}(0, \mathbf{I})$. For simplicity, by defining $\alpha_t = 1 - \beta_t$ and $\overline{\alpha}_t = \prod_{i=1}^{t} \alpha_i$, the diffusion process can be formulated as $q(\mathbf{x}_t|\mathbf{x}_0) = \mathcal{N}(\mathbf{x}_t; \sqrt{\overline{\alpha}_t}\mathbf{x}_0, (1 - \overline{\alpha}_t)\mathbf{I})$. Then in the generative process, a parameterized Markov chain is trained aiming to reverse the diffusion process and recover the image from the noise. To be specific, it learns model parameters $\theta$ such that the reverse transition $p_\theta(\mathbf{x}_{t-1}|\mathbf{x}_t)$, which is defined as $\mathcal{N}(\mathbf{x}_{t-1}; \boldsymbol{\mu}_\theta(\mathbf{x}_t, t), \boldsymbol{\Sigma}_\theta(\mathbf{x}_t, t))$, is equivalent to the forward transition $q(\mathbf{x}_{t-1}|\mathbf{x}_t, \mathbf{x}_0) = \mathcal{N}(\mathbf{x}_{t-1}; \hat{\boldsymbol{\mu}}_t(\mathbf{x}_t, \mathbf{x}_0), \hat{\beta}_t\mathbf{I})$, where $\hat{\boldsymbol{\mu}}_t(\mathbf{x}_t, \mathbf{x}_0) = \frac{1}{\sqrt{\alpha_t}}((\sqrt{\overline{\alpha}_t}\mathbf{x}_0 + (1 - \overline{\alpha}_t)\boldsymbol{\epsilon}) - \frac{\beta_t}{\sqrt{1-\overline{\alpha}_t}}\boldsymbol{\epsilon})$. To that end, DDPM aims to align the mean between $p_\theta(\mathbf{x}_{t-1}|\mathbf{x}_t)$ and $q(\mathbf{x}_{t-1}|\mathbf{x}_t, \mathbf{x}_0)$ via minimizing the following training objective:

$$\mathbb{E}_{t, \mathbf{x}_0, \boldsymbol{\epsilon}}[\left\|\boldsymbol{\epsilon} - \boldsymbol{\epsilon}_\theta(\sqrt{\overline{\alpha}_t}\mathbf{x}_0 + \sqrt{1 - \overline{\alpha}_t}\boldsymbol{\epsilon}, t)\right\|^2], \tag{1}$$

where $t$ is uniformly sampled from $\{1, \cdots, T\}$. Here, $\boldsymbol{\epsilon}_\theta$ represents a parameterized denoiser to predict noise $\boldsymbol{\epsilon}$ from $\mathbf{x}_t$, which is usually implemented based on U-Net Ronneberger et al. (2015).

### 3.2 Backdoor Attack on Diffusion Model

**Threat Model.** Following the settings in Chen et al. (2023); Chou et al. (2023), which assumes that the attacker can control the training process of a backdoored diffusion model, which will 1) generate the clean

image from the distribution $q(\mathbf{x}_0)$ with benign Gaussian noise input $\mathcal{N}(0, \mathbf{I})$; and 2) generate the target image from the distribution $\tilde{\mathbf{x}}_0 \sim \tilde{q}(\tilde{\mathbf{x}}_0)$ with the presence of poisoned noise input $\tilde{\mathbf{x}}_T$ that is embedded with a pre-defined *trigger* $\boldsymbol{\delta}$. Without loss of generality, we assume the trigger is proportionally blended to the clean Gaussian noise with propositional factor $\gamma \in [0, 1]$. More specifically, $\tilde{\mathbf{x}}_T \sim \mathcal{N}(\boldsymbol{\mu_\delta}, \gamma^2 \mathbf{I})$, where $\boldsymbol{\mu_\delta} = (1 - \gamma)\boldsymbol{\delta}$ satisfying $\tilde{\mathbf{x}}_T = (1 - \gamma)\boldsymbol{\delta} + \gamma\boldsymbol{\epsilon}, \boldsymbol{\epsilon} \sim \mathcal{N}(0, \mathbf{I})$. While this threat model might currently seem impractical, this series of works provides a valuable exploration for investigating the foundational property of backdoor attacks on the diffusion model. Following these works, we adopt this threat model in this paper.

**Backdoored Diffusion and Generative Processes.** To realize the attack goal, we assume that the adversary is allowed to modify the diffusion and generative processes and the training procedure. More specifically, as indicated in Chen et al. (2023), the attacker first diffuses the distribution $\tilde{q}(\tilde{\mathbf{x}}_0)$ of the target images to $\mathcal{N}(\boldsymbol{\mu_\delta}, \gamma^2 \mathbf{I})$, forming a backdoored diffusion process as $\tilde{q}(\tilde{\mathbf{x}}_{t-1} | \tilde{\mathbf{x}}_t, \tilde{\mathbf{x}}_0) = \mathcal{N}(\tilde{\mathbf{x}}_{t-1}; \tilde{\boldsymbol{\mu}}_t(\tilde{\mathbf{x}}_t, \tilde{\mathbf{x}}_0), \tilde{\beta}_t \mathbf{I})$, where $\tilde{\boldsymbol{\mu}}_t(\tilde{\mathbf{x}}_t, \tilde{\mathbf{x}}_0) = \frac{1}{\sqrt{\alpha_t}}((\sqrt{\overline{\alpha}_t}\tilde{\mathbf{x}}_0 + \sqrt{1 - \overline{\alpha}_t}\boldsymbol{\mu_\delta} + \sqrt{1 - \overline{\alpha}_t}\gamma\boldsymbol{\epsilon}) - \frac{\beta_t}{\sqrt{1 - \overline{\alpha}_t}}\boldsymbol{\epsilon})$. Then in the generative process, the parameterized model $\theta$ is learned to reverse both the benign and backdoored diffusion processes: $p_\theta(\mathbf{x}_{t-1} | \mathbf{x}_t) = q(\mathbf{x}_{t-1} | \mathbf{x}_t, \mathbf{x}_0)$ (for benign Gaussian input case described in Sec. 3.1) and $\tilde{p}_\theta(\tilde{\mathbf{x}}_{t-1} | \tilde{\mathbf{x}}_t) = \mathcal{N}(\tilde{\mathbf{x}}_{t-1}; \tilde{\boldsymbol{\mu}}_\theta(\tilde{\mathbf{x}}_t, t), \tilde{\boldsymbol{\Sigma}}_\theta(\tilde{\mathbf{x}}_t, t)) = \tilde{q}(\tilde{\mathbf{x}}_{t-1} | \tilde{\mathbf{x}}_t, \tilde{\mathbf{x}}_0)$ (for poisoned noise input case). To that end, the corresponding training objectives aim to simultaneously optimize both the benign and backdoor diffusion processes. Specifically, the benign training objective follows Sec. 1, and the backdoored diffusion training objective is formulated as:

$$\mathbb{E}_{t, \tilde{\mathbf{x}}_0, \boldsymbol{\epsilon}}[\|\boldsymbol{\epsilon} - \boldsymbol{\epsilon}_\theta(\sqrt{\overline{\alpha}_t}\tilde{\mathbf{x}}_0 + \sqrt{1 - \overline{\alpha}_t}\boldsymbol{\mu_\delta} + \sqrt{1 - \overline{\alpha}_t}\gamma\boldsymbol{\epsilon}, t)\|^2], \tag{2}$$

where $\tilde{\mathbf{x}}_0 \sim \tilde{q}(\tilde{\mathbf{x}}_0)$. Here $t, \boldsymbol{\epsilon}, \boldsymbol{\epsilon}_\theta$ are with the same setting in Eq. 1.

# 4 Trigger Detection in Backdoored Diffusion

Sec. 3.2 shows the feasibility of the backdoor attack on diffusion models via properly diffusing the target distribution and learning to reverse the backdoor generative process. Following this philosophy, some recent works Chen et al. (2023); Chou et al. (2023) have successfully launched the attack and demonstrated the vulnerability of the backdoored diffusion models with the presence of trigger patterns. However, we argue that the *detectability*, as an important attack performance metric, is not fully considered in the existing studies. More specifically, the embedded trigger patterns used in the state-of-the-art diffusion model backdoor attacks can be effectively detected.

Our key finding is that the poisoned noise containing the backdoor trigger can be distinguished from the clean Gaussian noise from the lens of data distribution.

As illustrated in Fig. 2, the distributions of two Gaussian noise inputs are highly overlapped; while an obvious distribution shift can be identified when comparing the poisoned noise and the benign Gaussian noise (see Fig. 2). Such phenomenon implies that the distribution discrepancy between the input and Gaussian noise $\mathcal{N}(0, \mathbf{I})$ can serve as a good marker to detect whether the input is potentially stamped with the backdoor trigger or not. To quantitatively measure this discrepancy, we propose to define a KL divergence-based KULLBACK (1959) Poisoned Distribution Discrepancy (PDD) score as follows:

$$D(\tilde{\mathbf{x}}_T) = \mathtt{KL}(P_{h(\tilde{\mathbf{x}}_T)}, P_{h(\mathbf{x}_T)}), \tag{3}$$

where $\mathbf{x}_T$ and $\tilde{\mathbf{x}}_T$ are the clean Gaussian input and the potentially poisoned input, respectively. $h(\cdot)$ denotes the histogram function, $P_{h(\cdot)}$ normalizes the histogram into a probability distribution, and $\mathtt{KL}(\cdot, \cdot)$ calculates KL divergence.

In general, for each potentially poisoned input, we can calculate its PDD score to evaluate its distribution shift from the benign Gaussian noise input. Notice that since even two clean Gaussian noises sampled from the same distribution still have a certain level of distribution discrepancy, such inevitable "base difference" incurred by the sampling randomness should be considered, and hence it can be empirically calculated as follows:

$$\phi_{Base} = \mathbb{E}_{\mathbf{x}_T}[D(\mathbf{x}_T)] + 3\sigma_{D(\mathbf{x}_T)}, \tag{4}$$

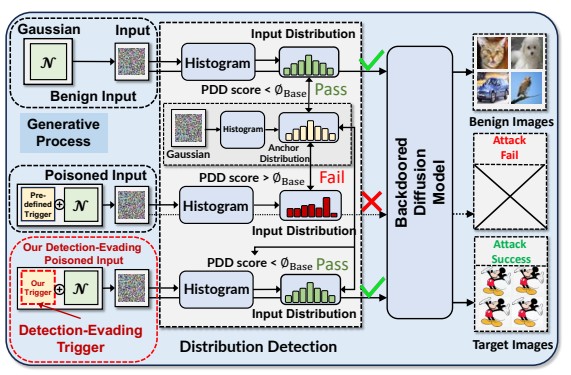

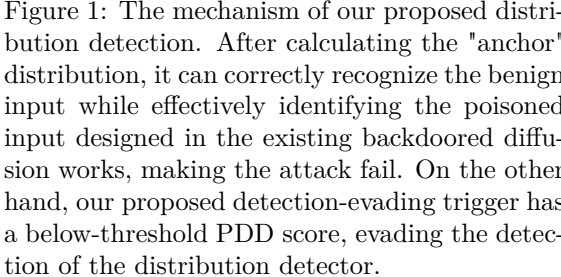

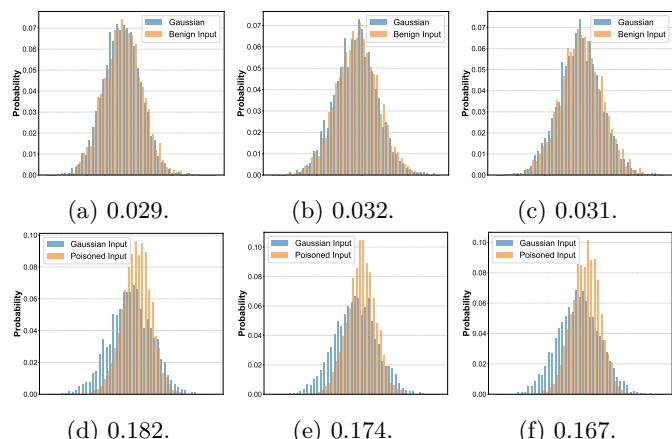

(a) 0.029.   (b) 0.032.   (c) 0.031.

(d) 0.182.   (e) 0.174.   (f) 0.167.

Figure 1: The mechanism of our proposed distribution detection. After calculating the "anchor" distribution, it can correctly recognize the benign input while effectively identifying the poisoned input designed in the existing backdoored diffusion works, making the attack fail. On the other hand, our proposed detection-evading trigger has a below-threshold PDD score, evading the detection of the distribution detector.

Figure 2: Distribution overlap between Gaussian noise and input noise. *(Top)*: Clean noise input (also Gaussian). *(Bottom)*: Poisoned noise input containing Hello Kitty trigger in Chen et al. (2023). It is seen that poisoned noise in the prior work exhibits a non-negligible distribution shift, bringing much higher PDD score than benign input.

where $\mathbf{x}_T^1, \mathbf{x}_T^2, \cdots, \mathbf{x}_T^N$ are the collection of clean inputs sampled from Gaussian distribution $\mathcal{N}(0, \mathbf{I})$. Also, considering the potential impact of statistical error on false positive rate, the calculation of base discrepancy includes an extra tolerance term (empirically set as $3\sigma_{D(\mathbf{x}_T)}$, more discussion to $X\sigma$ is in Sec. 7), ensuring that most ($> 99.8\%$) clean Gaussian noise inputs can be correctly recognized. Then, we can use this base discrepancy as the threshold to detect the backdoor trigger as follows:

**PDD-based Trigger Detection.** *Given an input noise $\tilde{\mathbf{x}}_T$, it will be detected as poisoned with backdoor trigger if $D(\tilde{\mathbf{x}}_T) \geq \phi_{Base}$; otherwise it is marked as clean.*

Fig. 1 illustrates the overall mechanism of the proposed distribution-based trigger detection approach. By preparing a set of clean Gaussian noise to compute the "anchor" distribution $P_{h(\mathbf{x}_T)}$ and base discrepancy $\phi_{Base}$ as the threshold, the detector can identify the poisoned noise input. As reported in our empirical evaluations (see Tab. 1), examining distribution shift shows very strong performance for detecting backdoor triggers. Our detection scheme only requires a one-time KL-divergence calculation between anchor and input distribution. For detecting a single input of the backdoored CIFAR-10 diffusion model, our method only needs 0.0014s, demonstrating its low-cost property.

## 5 Detection-Evading Backdoor Trigger Design

Sec. 4 analyzes the unique characteristics of the backdoor triggers for the diffusion models, and then develops the corresponding detection method. To deepen our understanding, in this section we further study the vulnerability of backdoored diffusion models from the perspective of attackers, exploring stealthy trigger design to evade the distribution-based detection mechanism.

### 5.1 Mitigate Distribution Discrepancy

As analyzed in Sec. 4, embedding the trigger to the benign Gaussian noise brings the detectable distribution shift. Therefore, in order to make the backdoor trigger undetectable, the PDD score of the poisoned noise $\tilde{\mathbf{x}}_T$ should be optimized and suppressed below the base discrepancy $\phi_{Base}$ as follows:

$$\max_{\boldsymbol{\delta}} \quad \frac{1}{N} \sum_{i=1}^{i=N} \mathbf{1}(D(\tilde{\mathbf{x}}_T^i) \leq \phi_{Base}), \tag{5}$$

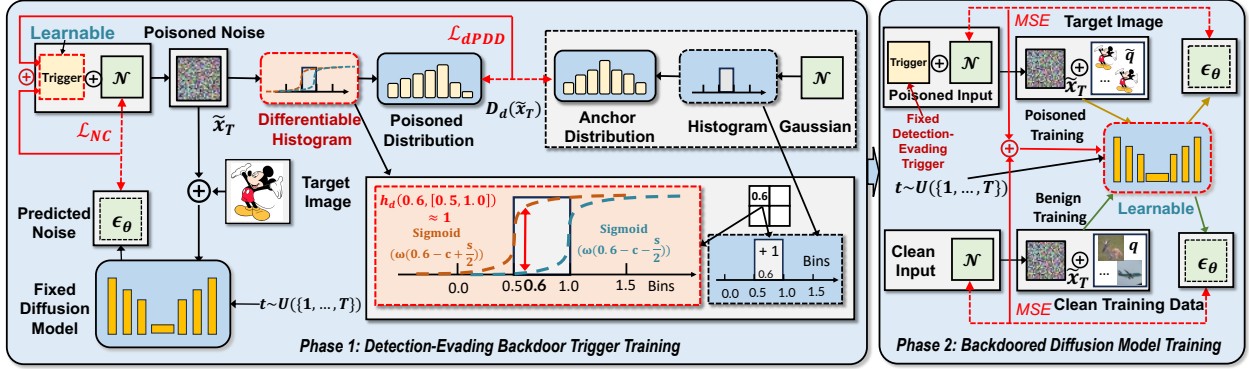

Figure 3: Our proposed two-step training scheme to learn the detection-evading trigger and the corresponding backdoored diffusion model. **Phase 1 (Left)**: Trigger is optimized by PDD loss $\mathcal{L}_{dPDD}$ and NC loss $\mathcal{L}_{NC}$ with the fixed diffusion model. To incorporate an end-to-end training procedure, we utilize the differentiable histogram $h_d(\cdot)$ for calculating $\mathcal{L}_{dPDD}$. **Phase 2 (Right)**: After optimizing the trigger, the diffusion model is updated towards the backdoored training objective with this detection-evading trigger.

where $\mathbf{1}(\cdot)$ is the indicator function, and $\tilde{\mathbf{x}}_T^i$ is one sample of poisoned noise input. Notice that here in order to mitigate the sampling error, the optimization of the backdoor trigger $\boldsymbol{\delta}$ is based on the evaluation of embedding $\boldsymbol{\delta}$ to $N$ benign Gaussian noise inputs $\mathbf{x}_T^i$. Then, the trigger can be learned via minimizing the following *PDD loss*:

$$\mathcal{L}_{PDD}(\boldsymbol{\delta}) = \mathbb{E}_{\tilde{\mathbf{x}}_T}[\max(D(\tilde{\mathbf{x}}_T) - \phi_{Th}, 0)], \tag{6}$$

where $\phi_{Th}$ is a pre-set threshold even smaller than $\phi_{Base}$, ensuring that after the training, the optimized PDD score can be optimized as being below $\phi_{Base}$ in a very probable way.

**Make Trigger Learning Differentiable.** In general, optimizing the PDD loss can be realized by using a gradient-based method such as stochastic gradient descent (SGD). However, as described in Eq. 3, the calculation of PDD score $D(\tilde{\mathbf{x}}_T)$ is involved with the non-differentiable histogram function $h(\cdot)$, preventing the differentiable learning of stealthy trigger. To address this problem, we propose to approximate the original histogram function to a differentiable format $h_d(\cdot)$. Here, the key idea is to use the dual logistic functions as a closed surrogate for the histogram function (see Fig. 3). More specifically, the differentiable histogram is calculated as:

$$h_d(\mathbf{x}, B_i) = \sum_{\mathbf{x}}((1 + e^{-\omega(\mathbf{x}-c_i+\frac{s_i}{2})})^{-1} - (1 + e^{-\omega(\mathbf{x}-c_i-\frac{s_i}{2})})^{-1}), \tag{7}$$

where $\omega$ controls the smoothness of the histogram, $B_i$ denotes the $i$-th bin in the histogram, and $c_i$ and $s_i$ represent the center and width of each bin, respectively. Then, the differentiable version of PDD score $D_d(\cdot)$ and loss used can be calculated as:

$$D_d(\tilde{\mathbf{x}}_T) = \text{KL}(P_{h_d(\tilde{\mathbf{x}}_T)}, P_{h_d(\mathbf{x}_T)}), \mathcal{L}_{dPDD}(\boldsymbol{\delta}) = \mathbb{E}_{\tilde{\mathbf{x}}_T}[\max(D_d(\tilde{\mathbf{x}}_T) - \phi_{Th}, 0)]. \tag{8}$$

**Two-Step Learning Procedure.** With the availability of differentiable PDD loss, the backdoored diffusion model and the corresponding detection-evading can be learned in an end-to-end manner. As shown in Fig. 3, we first fix the to-be-backdoored diffusion model and optimize the trigger by using PDD loss and NC loss (described in Sec. 5.2). After obtaining the stealthy trigger exhibiting low distribution discrepancy, we then fix this trigger and use it to generate poisoned input noise, facilitating the poison training for the backdoored diffusion model.

## 5.2 Noise Consistency Optimization

As shown in Fig. 3, in the trigger training phase, the *noise consistency loss* ($\mathcal{L}_{NC}$), which measures the discrepancy between the benign Gaussian noise input and the predicted noise $\boldsymbol{\epsilon}_\theta$, is also used to guide the

optimization of backdoor trigger $\boldsymbol{\delta}$. More specifically, the NC loss is defined and calculated as follows:

$$\mathcal{L}_{NC}(\boldsymbol{\delta}, \tilde{\mathbf{x}}_0) = \mathbb{E}_{t, \tilde{\mathbf{x}}_0, \boldsymbol{\epsilon}}[\|\boldsymbol{\epsilon} - \boldsymbol{\epsilon}_\theta(\sqrt{\overline{\alpha}_t}\tilde{\mathbf{x}}_0 + \sqrt{1 - \overline{\alpha}_t}\boldsymbol{\mu_\delta} + \sqrt{1 - \overline{\alpha}_t}\gamma\boldsymbol{\epsilon}, t)\|^2], \tag{9}$$

where $\tilde{\mathbf{x}}_0, t, \boldsymbol{\epsilon}, \boldsymbol{\epsilon}_\theta$ are with the same setting in Eq. 2. Here, the use of NC loss is motivated by the following design philosophy: Because the backdoored model training process (Phase 2) will use the exactly same discrepancy to update the learnable model (see Fig. 3), pre-optimize this loss in the trigger learning phase can provide better initialization and hence potentially improve both benign and attack performance. To be specific, the NC loss in the trigger training stage ensures that the trigger adapts well to the near-pretrained diffusion model. To maintain strong benign performance in the diffusion model, the training principle is to minimize weight updates relative to the pre-trained model. This regularized trigger effectively activates the backdoor in models that remain closely aligned with the pre-trained diffusion model. Without NC loss, the trigger lacks regularization and does not consider benign performance. The diffusion model would require significant modifications to achieve the attack objective, ultimately compromising its benign performance. With lower discrepancy between $\boldsymbol{\epsilon}$ and $\boldsymbol{\epsilon}_\theta$ for the poisoned training part, slight update from the original benign model may be already sufficient for fitting poisoned data samples, and hence the updated model, which is backdoored but closed to the original benign one, can probably perform well with the presence of benign inputs. Meanwhile, with the lower NC loss as the initialization, it is more likely to bring the poisoned training Fig. 3 to a better-optimized point after the same number of epochs, thereby improving the attack performance with the poisoned noise inputs. Notice that such a hypothesis has been verified in our empirical evaluations reported in Sec. 7. Algorithm 1 describes the overall 2-step training procedure, including using NC loss.

---

**Algorithm 1** The Proposed 2-Step Training Scheme

---

**Input:** Clean dataset $q(\mathbf{x}_0)$, backdoor target dataset $\tilde{q}(\tilde{\mathbf{x}}_0)$, pre-trained benign diffusion model $\theta$, scaling factor $\tau$, propositional factor $\gamma$, threshold $\phi_{Th}$, trigger learning rate $\eta_t$, model learning rate $\eta_d$.
**Output:** Detection-evading trigger $\boldsymbol{\delta}$, backdoored diffusion model $\theta_{bd}$.

1: *Phase 1: Detection-Evading Backdoor Trigger Training*
2: $\boldsymbol{\delta} \leftarrow \texttt{random}(\boldsymbol{\delta}.\texttt{shape})$
3: **repeat**
4:    $\tilde{\mathbf{x}}_0 \sim \tilde{q}(\tilde{\mathbf{x}}_0), t \sim \text{Uniform}(\{1, \cdots, T\}), \boldsymbol{\epsilon} \sim \mathcal{N}(0, \mathbf{I})$
5:    $\tilde{\mathbf{x}}_T \sim \mathcal{N}(\boldsymbol{\mu_\delta}, \gamma\boldsymbol{\epsilon})$                                      ▷ *Sample poisoned input noises*
6:    $\mathcal{L}_{dPDD}(\boldsymbol{\delta}) = \mathbb{E}_{\tilde{\mathbf{x}}_T}[\max(D_d(\tilde{\mathbf{x}}_T) - \phi_{Th}, 0)]$ via Eq. 8
7:    $\mathcal{L}_{NC}(\boldsymbol{\delta}; \tilde{\mathbf{x}}_0) = \mathbb{E}_{\tilde{\mathbf{x}}_0, t, \boldsymbol{\epsilon}}[\|\boldsymbol{\epsilon} - \boldsymbol{\epsilon}_\theta(\tilde{\mathbf{x}}_0, t, \boldsymbol{\delta})\|]$ via Eq. 9
8:    $\mathcal{L}(\boldsymbol{\delta}; \tilde{\mathbf{x}}_0) = \mathcal{L}_{NC}(\boldsymbol{\delta}; \tilde{\mathbf{x}}_0) + \tau\mathcal{L}_{dPDD}(\boldsymbol{\delta})$                       ▷ *Overall trigger loss*
9:    $\boldsymbol{\delta} \leftarrow \boldsymbol{\delta} - \eta_t \nabla_{\boldsymbol{\delta}} \mathcal{L}(\boldsymbol{\delta}; \tilde{\mathbf{x}}_0)$                                 ▷ *Updating trigger* $\delta$
10: **until** converged
11: *Phase 2: Backdoored Diffusion Model Training*
12: $\theta_{bd} \leftarrow \theta$                                 ▷ *Loading pre-trained benign diffusion model*
13: **repeat**
14:    $\mathbf{x}_0 \sim q(\mathbf{x}_0), t \sim \text{Uniform}(\{1, \cdots, T\}), \boldsymbol{\epsilon} \sim \mathcal{N}(0, \mathbf{I})$
15:    $\tilde{\mathbf{x}}_0 \sim \tilde{q}(\tilde{\mathbf{x}}_0), \tilde{t} \sim \text{Uniform}(\{1, \cdots, T\}), \tilde{\boldsymbol{\epsilon}} \sim \mathcal{N}(0, \mathbf{I})$
16:    $\mathcal{L}_c(\theta_{bd}) = \mathbb{E}_{\mathbf{x}_0, t, \boldsymbol{\epsilon}}[\|\boldsymbol{\epsilon} - \boldsymbol{\epsilon}_{\theta_{bd}}(\mathbf{x}_0, t, \boldsymbol{\epsilon})\|]$ via Eq. 1          ▷ *Benign*
17:    $\mathcal{L}_p(\theta_{bd}) = \mathbb{E}_{\tilde{\mathbf{x}}_0, \tilde{t}, \tilde{\boldsymbol{\epsilon}}}[\|\boldsymbol{\epsilon} - \boldsymbol{\epsilon}_{\theta_{bd}}(\tilde{\mathbf{x}}_0, \tilde{t}, \tilde{\boldsymbol{\epsilon}}, \boldsymbol{\delta})\|]$ via Eq. 2         ▷ *Poison*
18:    $\mathcal{L}(\theta_{bd}) = \mathcal{L}_c(\theta_{bd}) + \mathcal{L}_p(\theta_{bd})$                       ▷ *Backdoored model loss*
19:    $\theta_{bd} \leftarrow \theta_{bd} - \eta_d \nabla_{\theta_{bd}} \mathcal{L}(\theta_{bd})$                        ▷ *Updating diffusion model*
20: **until** converged

---

# 6 Experiments

**Datasets, Models and Attack Setting.** We evaluate the performance of the proposed detection method and the detection-evading trigger for DDPM Ho et al. (2020) and DDIM Song et al. (2020a) diffusion models on CIFAR-10 ($32 \times 32$) Krizhevsky et al. (2009) and CelebA ($64 \times 64$) Liu et al. (2015) datasets. The pre-trained models of CIFAR-10 and CelebA datasets are from repository `pesserpytorch/diffusion` and `ermongroup/ddim`. Two types of backdoor attack models are considered in the experiments: generate an image belonging to a specific class (referred to as "category mode") and generate a specific image (referred to as "instance mode"). Following the settings in Chen et al. (2023), we choose the horse in the CIFAR-10 dataset and faces with heavy makeup, mouth slightly open and smiling in the CelebA dataset as the target

Table 1: The effectiveness of the proposed distribution-based detection method for detecting the backdoor trigger used in Chen et al. (2023). The PDD score $D_d(\tilde{\mathbf{x}}_T) \gg \phi_{Base} = 0.067$ (for CIFAR-10 dataset) and $D_d(\tilde{\mathbf{x}}_T) \gg \phi_{Base} = 0.016$ (for CelebA dataset), **making the detection rate reach to 100% and ASR drop to 0%**, and Benign Pass Rate can reach 99.9%, indicating that benign generation is virtually unaffected.

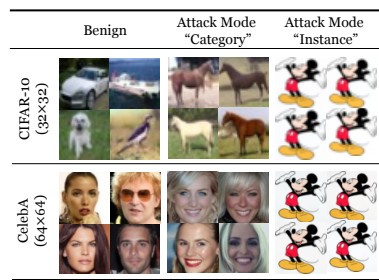

Figure 4: Generated images from our backdoored diffusion model. For CIFAR-10, the target class is "horse"; for CelebA, the target class includes faces characterized by "heavy makeup, smiling, and a slightly open mouth". The target image is a "Michy Mouse".

| Attack Mode | Average PDD Score | Benign Pass Rate (%) | Detection Rate (%) | DDPM ASR (%) w/o Detection | DDPM ASR (%) w/ Detection | DDIM ASR (%) w/o Detection | DDIM ASR (%) w/ Detection |
|---|---|---|---|---|---|---|---|
| CIFAR-10 | | | | | | | |
| Category | 0.183 | **99.9** | **100.0** | 90.1 | **0.0** | 87.30 | **0.0** |
| Instance | 0.183 | **99.9** | **100.0** | 100.0 | **0.0** | 100.0 | **0.0** |
| CelebA | | | | | | | |
| Category | 0.165 | **99.9** | **100.0** | 96.9 | **0.0** | 95.4 | **0.0** |
| Instance | 0.165 | **99.9** | **100.0** | 100.0 | **0.0** | 100.0 | **0.0** |

Table 2: Performance of the proposed backdoored diffusion model using the detection-evading learnable trigger. The lower FID and higher ASR indicate better benign and attack performance, respectively. For "Instance" mode, MSE between the generated and target images is also reported to show the high attack performance. "∗" denotes we reproduce the experiments using "Glasses" Chou et al. (2023) as the trigger and "Micky Mouse" as the target image.

| Attack Mode | Method | Trigger Type | PDD Score | Detection Pass Rate (%) | DDPM Benign FID ↓ | DDPM Benign Δ FID | DDPM Attack ASR (%) w/ Detection ↑ | DDIM Benign FID ↓ | DDIM Benign Δ FID | DDIM Attack ASR (%) w/ Detection ↑ |
|---|---|---|---|---|---|---|---|---|---|---|
| CIFAR-10 | | | | | | | | | | |
| None | Benign Baseline | None | 0.031±0.012 | 99.8 | 4.60 | 0 | 0 | 4.25 | 0 | 0 |
| Category | Trojdiff Chen et al. (2023) | Fixed | 0.183±0.012 | 0.0 | 4.74 | 0.14 | 0.0 | 4.47 | 0.22 | 0.0 |
| | **DisDet(Ours)** | **Learnable** | **0.025±0.007** | **99.9** | **4.44** | **-0.16** | **82.0** | **4.29** | **0.04** | **80.1** |
| Instance | Trojdiff Chen et al. (2023) | Fixed | 0.183±0.012 | 0.0 | 4.59 | -0.01 | 0.0 | 4.47 | 0.22 | 0.0 |
| | Baddiffusion∗ Chou et al. (2023) | Fixed | 0.269±0.015 | 0.0 | 4.52 | -0.08 | 0.0 | 4.43 | 0.18 | 0.0 |
| | **DisDet(Ours)** | **Learnable** | **0.025±0.007** | **99.9** | **4.39** | **-0.21** | **99.9 (MSE: 7.64e-6)** | **4.38** | **0.13** | **99.9 (MSE: 4.19e-5)** |
| CelebA | | | | | | | | | | |
| None | Benign Baseline | None | 0.007±0.003 | 99.8 | 5.88 | 0 | 0 | 6.29 | 0 | 0 |
| Category | Trojdiff Chen et al. (2023) | Fixed | 0.165±0.006 | 0.0 | 5.44 | -0.44 | 0.0 | 5.40 | -0.89 | 0.0 |
| | **DisDet(Ours)** | **Learnable** | **0.007±0.003** | **99.8** | **5.83** | **-0.05** | **85.9** | **5.94** | **-0.35** | **85.2** |
| Instance | Trojdiff Chen et al. (2023) | Fixed | 0.165±0.006 | 0.0 | 5.62 | -0.26 | 0.0 | 5.93 | -0.36 | 0.0 |
| | Baddiffusion∗ Chou et al. (2023) | Fixed | 0.260±0.007 | 0.0 | 5.73 | -0.15 | 0.0 | 5.98 | -0.31 | 0.0 |
| | **DisDet(Ours)** | **Learnable** | **0.007±0.003** | **99.8** | **5.80** | **-0.08** | **99.8 (MSE: 1.52e-3)** | **5.85** | **-0.44** | **99.8 (MSE: 1.70e-3)** |

class in the category mode. The Micky Mouse is selected as the target image when the backdoor attack is launched in the instance mode. The "Hello Kitty" and "Glass" images are set as fixed triggers in experiments of TrojDiff Chen et al. (2023) and Baddiffusion Chou et al. (2023), respectively.

**Training Configurations.** When training the detection-evading trigger (Phase 1), an `Embedding` layer with the same shape of input noise ($3 \times 32 \times 32$ for CIFAR-10 dataset, and $3 \times 64 \times 64$ for CelebA dataset) is used for trigger learning with $\gamma = 0.6$. The threshold is set as $\phi_{Th} = 0.01$ and $\phi_{Th} = 0.005$ for the CIFAR-10 and CelebA datasets, respectively. The training process adopts Adam optimizer Kingma & Ba (2015) with 50k training steps, $2 \times 10^{-3}$ learning rate and scaling factor $\tau$ as $10^4$. After that, during the training procedure for the backdoored diffusion model (Phase 2), we follow the standard training procedure using Adam optimizer, $2 \times 10^{-4}$ learning rate, batch size as 256, and 100k training steps. Also, the number of bins is set as 50 for both regular histogram $h(\cdot)$ and differentiable histogram $h_d(\cdot)$. The smoothness parameter is set as $\omega = 6$ for the Sigmoid function in $h_d(\cdot)$ to approximate the step function and histogram $h(\cdot)$. All the experiments are conducted on NVIDIA RTX A6000 GPUs.

**Evaluation Metrics.** The benign performance is evaluated on 50K samples via measuring Frechet Inception Distance (FID) Heusel et al. (2017), which reveals the similarity between two sets of images. A lower FID score indicates the higher quality of the generated images. The attack performance is evaluated on 10K samples in terms of Attack Success Rate (ASR). When the attack mode is set as "category" and "instance", ASR is measured as the ratio of the generated images being classified into the target class and being the same as the target image, respectively. Specifically, when the attack is launched in the "instance" mode, we also measure the Mean Square Error (MSE) to examine the difference between the target image and the generated images. For the image sampling, we follow the standard strategy by setting $\eta = 1$ and $S = 1000$ and $\eta = 0$ and $S = 100$ in DDPM Song et al. (2020a) and DDIM Ho et al. (2020), respectively.

## 6.1 Evaluation Results

**CIFAR-10 Dataset.** Tab. 2 shows the benign and attack performance on the CIFAR-10 dataset. When the backdoor attack is launched in the "category" mode, the average PDD score of the poisoned noise generated from the trigger designed in Chen et al. (2023) is 0.183. This score is significantly higher than the base discrepancy $\phi_{Base} = 0.067$, making the attack can be easily detected with ASR as 0. In contrast, our detection-evading trigger is learned to exhibit a very low PDD score of 0.025, making the attack very undetectable (nearly 100% detection pass rate) with high ASR (more than 80%). Meanwhile, it enjoys good benign performance with even lower FID than the baseline (originally non-backdoored case). In other words, with the presence of benign input, the images generated by our backdoored model have even higher quality than the ones generated by the diffusion model without backdoor injection. Similarly, in the backdoor attack mode "instance", our approach also shows much better benign and attack performance than the prior works. In particular, the MSE between the generated images and the original target image (Mickey Mouse) is very small (7.64e-6 and 4.19e-5 for DDPM and DDIM, respectively), indicating the effectiveness of the attack.

**CelebA Dataset.** As shown in Tab. 2, our optimized trigger is effectively stealthy to the distribution detector and achieves higher attack performance than the prior works. More specifically, with $\phi_{Base}$ as 0.016, the average PDD score of the poisoned noise in Chen et al. (2023) ( "category" attack mode) is 0.165, bringing an ASR of 0 since all the triggers will be detected. On the other hand, the average PDD score of our detection-evading trigger is only 0.007, and hence it is very undetectable to the distribution detector, bringing high ASR (more than 85%). Meanwhile, the benign performance of our solution is good with even lower FID than the baseline design. Similarly, in the 'instance" attack mode, our method enjoys a low average PDD score of 0.007 and also FID reduction as compared to the baseline case, demonstrating high attack and benign performance.

**Detection Results.** We have shown the detection results of "Baddiffusion" in the column "Detection Pass Rate" in Tab. 2. Similar to Trojdiff, our detection method can achieve a 100% detection rate for "Baddiffusion" attacks.

**Visualization.** Fig. 4 illustrates some of the generated images from our backdoored diffusion model with benign and poisoned noise inputs. It is seen that our approach is very effective in both benign and attack scenarios.

## 7 Ablation Studies

**Effect of PDD Optimization.** Fig. 5 shows the curve of differentiable PDD score $D_d(\tilde{\mathbf{x}}_T)$ during the trigger training procedure. It is seen that it steadily decreases as training progresses, and finally this loss reaches below $\phi_{Th}$, indicating that the proposed differentiable histogram $h_d(\cdot)$ is an effective approximation to $h(\cdot)$ when considering the gradient-based optimization. Also, Fig. 6 illustrates the distribution discrepancy incurred by fixed trigger used in Chen et al. (2023) and learnable trigger after PDD optimization. It is seen that the proposed PDD-oriented trigger learning brings a much lower PDD score, significantly improving the stealthiness of the backdoor trigger.

We also observe the variation of the absolute mean of the trigger values when optimizing the PDD scores, as illustrated in Fig. 7. Although the mean of trigger values remains around zero, there is a significant variation in absolute mean values. As the training process progresses, the absolute mean increases while the

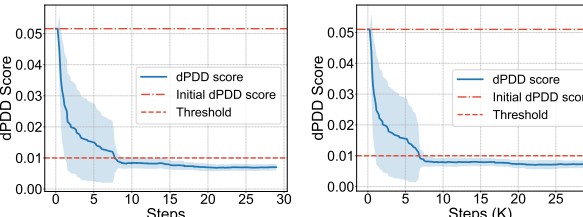

(a) Attack mode: "Category".

(b) Attack mode: "Instance".

Figure 5: Curve of differentiable PDD score $D_d(\tilde{\mathbf{x}}_T)$ when the trigger is trained with the PDD loss $\mathcal{L}_{dPDD}$ on the CIFAR-10 dataset. $D_d(\tilde{\mathbf{x}}_T)$ steadily decreases and reaches below threshold.

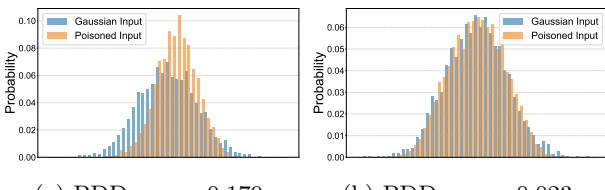

(a) PDD score: 0.179.

(b) PDD score: 0.023.

Figure 6: Distribution of the poisoned inputs with (a) the fixed trigger Chen et al. (2023) and (b) our proposed learnable trigger on CIFAR-10 dataset. PDD-oriented optimization brings a much lower PDD score, making our poisoned inputs approach clean Gaussian noise.

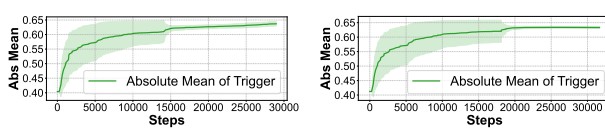

(a) Mode: "Category".

(b) Mode: "Instance".

Figure 7: The curve of absolute mean of trigger values during the PDD-oriented training process.

Table 3: FID results of backdoored diffusion model with the trigger optimized with or without NC optimization. The results are evaluated with different sampling steps on CIFAR-10 dataset.

| | Init Loss | Sampling Steps, $\eta = 0$ | | |
|---|---|---|---|---|
| | | 50 | 100 | 200 |
| w/o NC | 205.1 | 5.07 | 4.51 | 4.50 |
| **w/ NC** | **74.3** | **4.99** | **4.38** | **4.17** |
| $\Delta$ | **130.8** | **0.08** | **0.13** | **0.33** |

PDD score decreases. This phenomenon leads us to hypothesize that the change in the absolute mean value of the trigger may be related to the process of approaching Gaussian distribution. The modified values seem to mimic the distribution of Gaussian noise. As the PDD scores reach a plateau, the mean absolute value also becomes stable, no longer experiencing significant fluctuations.

Additionally, Wang et al. (2024) demonstrates that a larger magnitude enhances performance on both clean and backdoored data. Consistent with this finding, the magnitude of our trigger increases during PDD-oriented training, further supporting this observation and potentially offering new insights.

**Effect of NC Optimization.** As analyzed in Sec. 5.2, NC loss measures the discrepancy between the added Gaussian noise and the predicted noise at one step. Fig. 8 shows the curve of NC loss as training progresses. It is seen that compared with only emphasizing optimization of PDD, training towards both optimizing both PDD and NC brings a very significant NC loss drop, indicating the strong noise prediction capability and generation of higher-quality images. As shown in Tab. 3, using NC optimization leads to lower FID scores across different sampling steps. Fig. 9 visualizes the benign and attack performance improvement after considering NC optimization in trigger learning.

**Effect of Various Smoothness Factors.** To explore the effect of the smoothness factor $\omega$ of the differential histogram function $h_d(\cdot)$, we conduct several PDD optimization experiments using various $\omega$. Fig. 10 shows the curve of the PDD score $D(\tilde{\mathbf{x}}_T)$ and differentiable PDD score $D_d(\tilde{\mathbf{x}}_T)$ during the trigger PDD optimization procedure.

When $\omega$ is too small, such as $\omega = 2$ in Fig. 10a, the approximation of differentiable PDD score $D_d(\tilde{\mathbf{x}}_T)$ becomes less accurate. This inaccuracy results in a very small differential PDD score, even at the initialization step. Then, it leads to a larger gap between the regular PDD score $D(\tilde{\mathbf{x}}_T)$ and differentiable PDD score $D_d(\tilde{\mathbf{x}}_T)$. This larger gap is undesirable for optimizing the desired distribution pass rate and ASR. When $\omega$ is too large (e.g., $\omega = 8$), the differential histogram function exhibits non-smooth characteristics. This can be seen in Fig. 10d, where after 2.5K steps, the PDD scores become undefined (denoted as "NaN"). This phenomenon is directly attributed to the non-smooth nature of the differential histogram function with larger $\omega$ values. Consequently, a larger $\omega$ not only hinders but also halts the optimization process of the

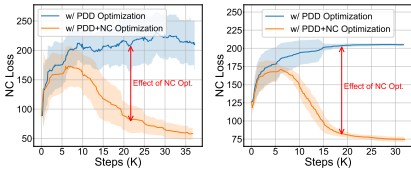

(a) Attack mode: "Category".  (b) Attack mode: "Instance".

Figure 8: Effect of NC optimization. $\mathcal{L}_{NC}$ in the y-axis denotes the deviation between the predicted and added noises of the diffusion model on CIFAR-10 dataset. Optimizing the trigger $\boldsymbol{\delta}$ with NC loss facilitates the model more easily predicting the noise in the poisoned input noise $\tilde{\mathbf{x}}_T$. $\mathcal{L}_{NC}$ in attack mode "instance" is stable since the target image is fixed, causing less variance.

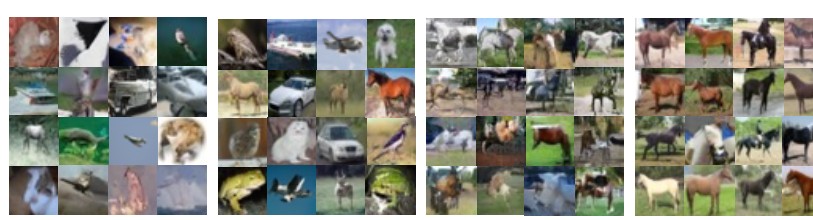

(a) w/ PDD, w/o NC.  (b) w/ PDD, w/ NC.  (c) w/ PDD, w/o NC.  (d) w/ PDD, w/ NC.

Figure 9: Visualization of generated CIFAR-10 images with or without NC optimization. Figures (a) and (b) display generated clean images with benign input (i.e., bird, ship, airplane, dog, horse, automobile, deer, frog). Figures (c) and (d) show generated images of the target class ("horse") with poisoned inputs. Figures (b) and (d) represent the images generated from poisoned inputs with the NC-optimized trigger. NC optimization is observed to enhance the image quality in both benign and attack scenarios.

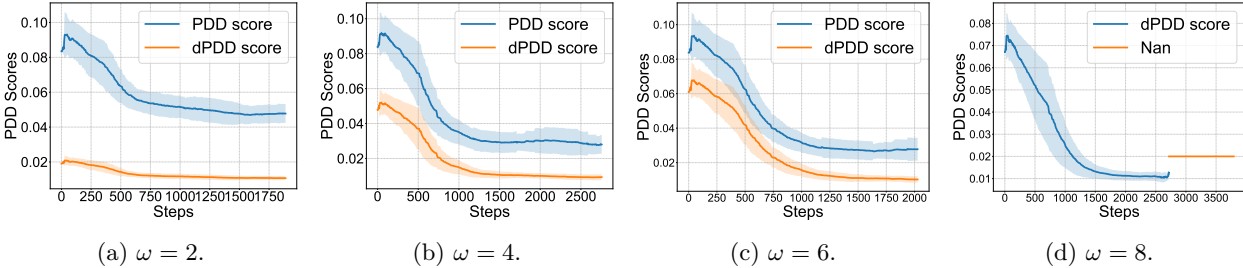

(a) $\omega = 2$.  (b) $\omega = 4$.  (c) $\omega = 6$.  (d) $\omega = 8$.

Figure 10: Curve of PDD score $D(\tilde{\mathbf{x}}_T)$ and differentiable PDD score $D_d(\tilde{\mathbf{x}}_T)$ when the trigger is trained with the PDD loss $\mathcal{L}_{dPDD}$. Effect of $\omega$, the smoothness factor of the differential histogram function.

desired PDD score. In our experiments, we choose $\omega = 6$ to strike a balance between minimizing the gaps in PDD score and maintaining the feasibility of optimization.

**Effect of Various $X\sigma$.** We follow the 3-sigma rule in Gaussian distribution to set $3\sigma$ in our main experiments because it provides a way to manage the spread of data within a predictable range. In a Gaussian distribution, about 99.7% of the data points lie within three standard deviations (3-sigma) from the mean to avoid hurting the pass rate of the benign input. Specifically, if the defender applies a stricter criterion, such as $1\sigma$, the detection system would significantly decrease the pass rate of benign inputs from 99.8% to 84%, thereby increasing the 20% number of generation attempts (number of trials). Therefore, to improve the user experience, we use $3\sigma$ as a default setting. Additionally, we conduct the experiment on the CIFAR-10 dataset and present the effect of various $X\sigma$ in Tab. 4. Our DisDet can achieve 97.5% ASR with the attack mode "Instance".

**Defend the patched-based trigger.** We conduct additional experiments incorporating a patch-based trigger to further validate the universality of our method. Based on the CIFAR-10 dataset, we use the detection method outlined in Sec. 4 to detect the patch trigger from Trojdiff. The results show that our PDD-based defense can achieve a 90.4% detection rate for the path-based trigger, resulting in a low ASR of 9.6%. This indicates that our detection method is effective for both the blend and patch-based trigger.

**Sampling Steps and Training Epochs.** To investigate how training and sampling steps affect the performance of a stealthy backdoor diffusion model, we conduct training on CIFAR-10 and CelebA datasets using the "instance" attack mode. The models are trained with varying training steps (50k, 75k, 100k) and

Table 4: Effect of various $X\sigma$. Performance of the proposed backdoored diffusion model using the detection-evading learnable trigger under $1\sigma$. The lower FID and higher ASR indicate better benign and attack performance, respectively. For "Instance" mode, MSE between the generated and target images is also reported to show the high attack performance. "$*$" denotes we reproduce the experiments using "Glasses" Chou et al. (2023) as the trigger and "Micky Mouse" as the target image.

| Attack Mode | Method | Trigger Type | PDD Score | Detection Pass Rate (%) | DDPM | | | DDIM | | |
|---|---|---|---|---|---|---|---|---|---|---|
| | | | | | Benign | | Attack | Benign | | Attack |
| | | | | | FID ↓ | Δ FID | ASR (%) w/ Detection ↑ | FID ↓ | Δ FID | ASR (%) w/ Detection ↑ |
| CIFAR-10 | | | | | | | | | | |
| None | Benign Baseline | None | 0.031±0.012 | 84.0 | 4.60 | 0 | 0 | 4.25 | 0 | 0 |
| Category | DisDet(Ours) | Learnable | 0.025±0.007 | 97.5 | 4.44 | -0.16 | 80.0 | 4.29 | 0.04 | 78.1 |
| Instance | DisDet(Ours) | Learnable | 0.025±0.007 | 97.5 | 4.39 | -0.21 | 97.5 (MSE: 7.64e-6) | 4.38 | 0.13 | 97.5 (MSE: 4.19e-5) |

sampling steps (10, 20, 50, 100, 200). According to the results presented in Tab. 5, an increase in training epochs generally maintains or slightly decreases the FID while simultaneously enhancing attack performance. Regarding sampling steps, we observe that larger number of sampling steps leads to improved performance in both benign and attack scenarios. This aligns with the property of a standard diffusion model, where increasing sampling steps tend to yield better overall performance.

Table 5: Results of different sampling steps on various training epochs on CIFAR-10 and CelebA datasets with attack mode "instance". For the benign performance, FID reflects the quality of benign images. For the attack performance, the ASR equals that of the main results, and MSE is measured to reflect the subtle change exactly according to different sampling steps and training epochs.

| Training Epochs | Metric | Sampling Steps | | | | |
|---|---|---|---|---|---|---|
| | | 10 | 20 | 50 | 100 | 200 |
| CIFAR-10 | | | | | | |
| 50k | FID | 14.74 | 7.42 | 5.00 | 4.39 | 4.17 |
| | MSE | 1.53e-4 | 1.03e-4 | 8.37e-5 | 8.10e-5 | 7.34e-5 |
| 75k | FID | 14.37 | 7.33 | 4.98 | 4.34 | 4.17 |
| | MSE | 1.07e-4 | 7.12e-5 | 5.69e-5 | 5.52e-5 | 5.02e-5 |
| 100k | FID | 14.51 | 7.38 | 4.99 | 4.38 | 4.17 |
| | MSE | 8.20e-5 | 5.43e-5 | 4.32e-5 | 4.19e-5 | 3.80e-5 |
| CelebA | | | | | | |
| 50k | FID | 2.85e-3 | 2.02e-3 | 1.71e-3 | 1.70e-3 | 1.59e-3 |
| | MSE | 13.31 | 8.23 | 6.35 | 5.85 | 5.75 |
| 75k | FID | 1.43e-3 | 1.04e-3 | 9.01e-4 | 8.98e-4 | 8.55e-4 |
| | MSE | 13.24 | 8.17 | 6.28 | 5.84 | 5.71 |
| 100k | FID | 1.08e-3 | 7.98e-4 | 7.01e-4 | 6.99e-4 | 6.69e-4 |
| | MSE | 13.25 | 8.14 | 6.23 | 5.82 | 5.72 |

**Other Techniques for Removing Backdoored Triggers.** Several other techniques aim to remove backdoor triggers rather than detect them, including Gaussian smoothing, denoising, filtering, purification, and reconstruction methods. Although these methods can defend against backdoor attacks, they negatively impact the quality of benign generation. We conduct the experiments with the attack mode "Instance" on CIFAR-10 with Gaussian smoothing (kernel size is 5) and Median Filter (window size is 5). As shown in Table 6, our detection method can keep better FID of generative benign images. Take the Gaussian smoothing as an example, suppose given a Gaussian noise $\{X_i\}_{i=1}^N$, where each $X_i \sim N(0, 1)$ and all $X_i$ are independent. Define the Gaussian-smoothed output as $Y_i = \sum_j K(i-j) X_j$,, where $K(\cdot)$ is a discrete Gaussian kernel. Each $Y_i$ is a linear combination of i.i.d. Gaussians, so $Y_i \sim N\left(0, \sigma_{Y_i}^2\right)$, with $\sigma_{Y_i}^2 = \sum_j K(i-j)^2$, assuming a normalized kernel over appropriate indices. For $i \neq m$, $\text{Cov}(Y_i, Y_m) = \sum_j \sum_k K(i-$

Table 6: Results of different trigger removing techniques.

| Methods | ASR | Benign FID |
|---|---|---|
| Trojdiff | 100% | 4.59 |
| Gaussian Smoothing | 5% | 30.86 |
| Median Filter | 6% | 37.15 |
| Ours | 0% | 4.63 |

$j) K(m - k) \operatorname{Cov}(X_j, X_k) = \sum_j K(i - j) K(m - j)$, since $\operatorname{Cov}(X_j, X_k) = 0$ for $j \neq k$ and $= 1$ for $j = k$. Hence $Y_i$ and $Y_m$ become correlated when their kernels overlap. As a result, the benign Gaussian noise is transformed into non-standard Gaussian noise, which degrades the quality of the generated images.

**Discussion of the trade-off between attack capability and stealthiness against our proposed detection method.** There exists a trade-off in previous methods, such as Trojdiff Chen et al. (2023), which can decrease the magnitude of the backdoor trigger to bypass the proposed defense mechanism. As the magnitude of the trigger in TrojDiff decreases, it becomes easier to evade distribution-based detection.

As a result, this also leads to a lower ASR, highlighting a trade-off between ASR and detection rate. Here we conduct the experiments and present some results in Table 7. In the experiments, we applied the PDD threshold as $\mathbb{E}[D]$ and adjusted the default magnitude in TrojDiff. Our results illustrate the trade-offs associated with different TrojDiff magnitude settings. Note that a higher magnitude factor corresponds to a smaller trigger magnitude. Furthermore, we also evaluate our proposed attack method with the same setting and same PDD threshold. The results demonstrate that our proposed attack consistently achieves a higher ASR.

Table 7: Trade-off between attack capability and stealthiness of Trojdiff Chen et al. (2023) against our proposed detection method. Note that a higher magnitude factor corresponds to a smaller trigger magnitude.

| Magnitude Factor | Pass Rate | ASR |
|---|---|---|
| Trojdiff-0.95 | 76.6% | 36.5% |
| Trojdiff-0.90 | 59.3% | 48.6% |
| Trojdiff-0.85 | 4.5% | 3.7% |
| Trojdiff-0.80 | 0% | 0% |
| Ours Attack | 80.0% | 64.0% |

## 8 Conclusion

In this paper, we perform studies on the detectability of the Trojan noise input for the backdoored diffusion model, from both defender and attacker sides. First, we propose the distribution discrepancy-based trigger detection mechanism for detecting the backdoor attack. Second, we design a detection-evading trigger to unnoticeably attack the diffusion model. Evaluation results on CIFAR-10 and CelebA datasets show a nearly 100% detection rate to the previous backdoor attack on diffusion models, and a 100% detection pass rate with very high attack and benign performance of our proposed attack method. Our attack method shows a vulnerability of the PDD score-based defense, and future approaches should consider an alternate framework instead. Note that although we provide the conventional filtering techniques to remove backdoored triggers in ablation studies, more complicated denoising or filtering methods might also achieve the same effect, such as incorporating statistical regularization during the training of the denoiser to maintain Gaussian characteristics, or utilizing probabilistic denoising methods. Therefore, more advanced filtering techniques remain underexplored, and these more complex methods could be conducted as a separate research work in the future.

## 9 Broader Impact

By exploring the detectability of poisoned noise input in backdoored diffusion models, this paper contributes significantly to improving the security of generative AI models. The proposed trigger detection mechanism can potentially safeguard these systems against backdoor attacks, ensuring that content generation tools remain trustworthy and reliable. The proposed backdoor attack strategy that learns to evade detection could pave the way for future research aimed at exploring and enhancing the robustness of generative AI models.

**Acknowledgments**

This work was supported in part by the National Science Foundation (NSF) under grants CCF2211163, IIS2311596, IIS2311597, CNS2114220, CNS2145389, CNS2350188.

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

## Appendix

## A    Limitations

The threat model in the previous backdoored diffusion model Chen et al. (2023); Chou et al. (2023) assumes that the attacker can control the training process and access the input Gaussian noise of the diffusion model. While these assumptions might currently seem impractical, this series of works provides a valuable exploration for investigating the foundational property of backdoor attacks on the diffusion model. Following these works, we adopt this threat model to guide our analysis. Second, our study currently focuses on unconditional diffusion models, in future work, we aim to enhance our understanding of detection mechanisms within conditional diffusion models, such as Stable Diffusion.

## B  Potential Scenarios

Here, we present potential scenarios the threat model could arise:

- Scenario 1: When developing the diffusion model on a device, the hardware could be poisoned by continuously shifting the input noise with a trigger. This would cause the model to generate dangerous and unpredictable content.

- Scenario 2: If a company allows users to input their own Gaussian noise via an open API to generate corresponding target images, users might save a fixed Gaussian noise that generates a desired image. This feature can attract customers to use the model from the company. However, malicious users could exploit this input method to introduce harmful triggers, resulting in dangerous outputs.

## C  Details of Differentiable Histogram

We propose to use the dual logistic functions to approximate the original histogram function. Given a bin with the range of $[0.5, 1.0]$, with the center $c = 0.75$ and interval $s = 0.5$, when an item with a value of 0.6 falls into this bin, the histogram count increases by 1. To make this process differentiable, we use dual logistic functions to approximate this count. As seen in Fig. 3, we use a brown dotted line to represent $\texttt{sigmoid}(\omega(x - c + \frac{s}{2}))$, corresponding to the first term of Eq. 7. The blue dotted line represents $\texttt{sigmoid}(\omega(x - c - \frac{s}{2}))$, corresponding to the second term in Eq. (7). In the figure, using $x = 0.6$ as an example, this count of 1 is approximated by subtracting the blue line value from the brown line value, which is denoted by $\texttt{sigmoid}(\omega(0.6 - 0.75 + \frac{0.5}{2})) - \texttt{sigmoid}(\omega(0.6 - 0.75 - \frac{0.5}{2}))$. By summing all the counts for each bin separately, Eq. 7 can approximate the histogram function.

## D  Additional Visualization Results

### D.1  CIFAR-10 Dataset

**DDPM.** Fig. 11, Fig. 12, Fig. 13, Fig. 14 illustrate some of the generated images from our backdoored DDPM model with benign and poisoned noise inputs on CIFAR-10 dataset. For the attack mode "Category", the target category is the horse. For the attack mode "Instance", the target instance is a Michy Mouse image.

**DDIM.** Fig. 15, Fig. 16, Fig. 17, Fig. 18 illustrate some of the generated images from our backdoored DDIM model with benign and poisoned noise inputs on CIFAR-10 dataset. For the attack mode "Category", the target category is the horse. For the attack mode "Instance", the target instance is a Michy Mouse image.

### D.2  CelebA Dataset

**DDPM.** Fig. 19, Fig. 20, Fig. 21, Fig. 22 illustrate some of the generated images from our backdoored DDPM model with benign and poisoned noise inputs on CelebA dataset. For the attack mode "Category", the target category is the faces with heavy makeup, mouth slightly open and smiling. For the attack mode "Instance", the target instance is a Michy Mouse image.

**DDIM.** Fig. 23, Fig. 24, Fig. 25, Fig. 26 illustrate some of the generated images from our backdoored DDIM model with benign and poisoned noise inputs on CelebA dataset. For the attack mode "Category", the target category is the faces with heavy makeup, mouth slightly open and smiling. For the attack mode "Instance", the target instance is a Michy Mouse image.

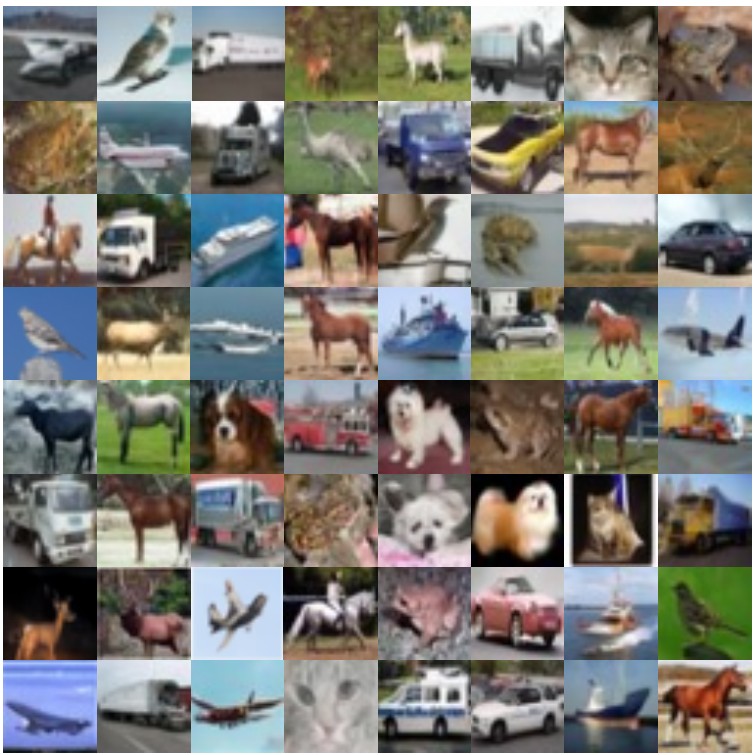

Figure 11: Generated benign images in Attack Mode "Category" with DDPM model on CIFAR-10 dataset.

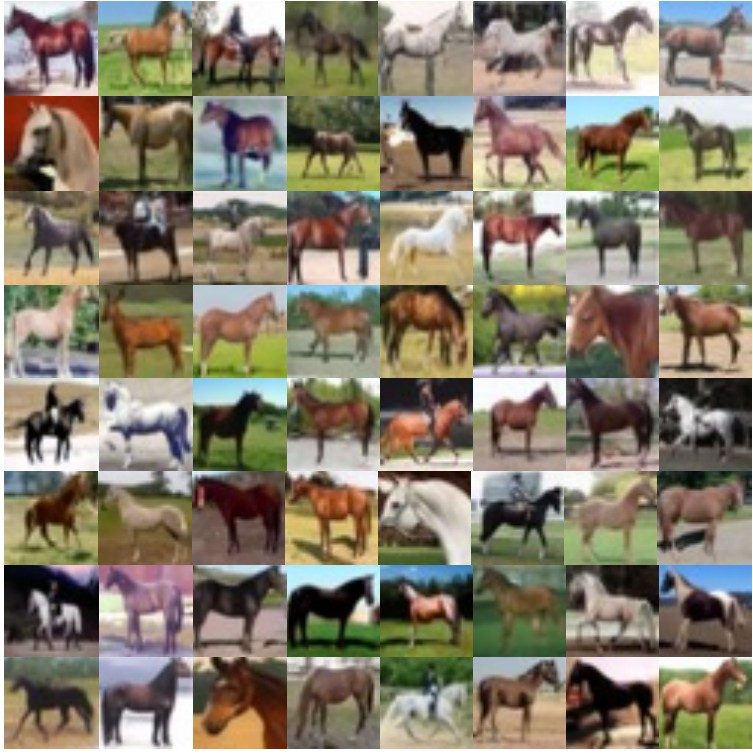

Figure 12: Generated target images in Attack Mode "Category" with DDPM model on CIFAR-10 dataset. The target category is the horse.

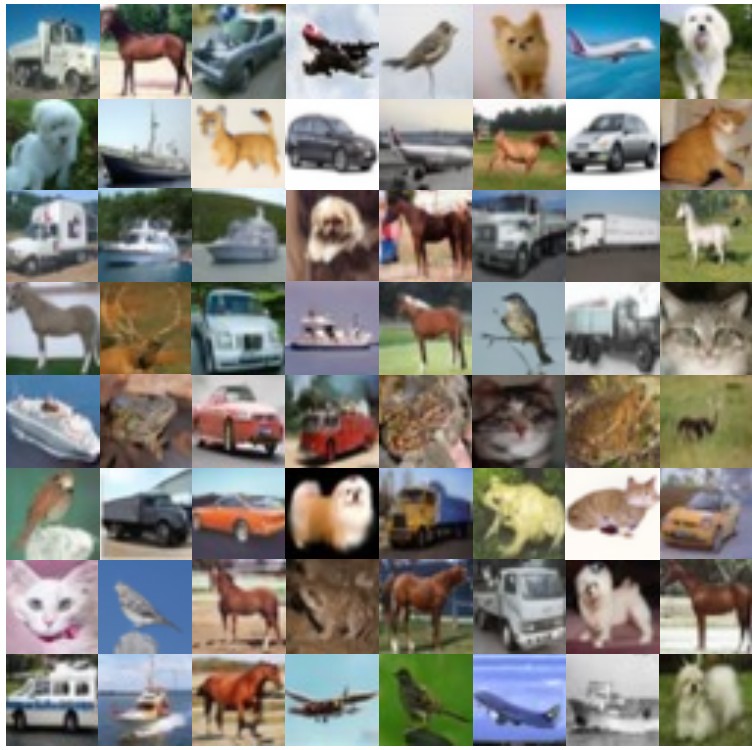

Figure 13: Generated benign images in Attack Mode "Instance" with DDPM model on CIFAR-10 dataset.

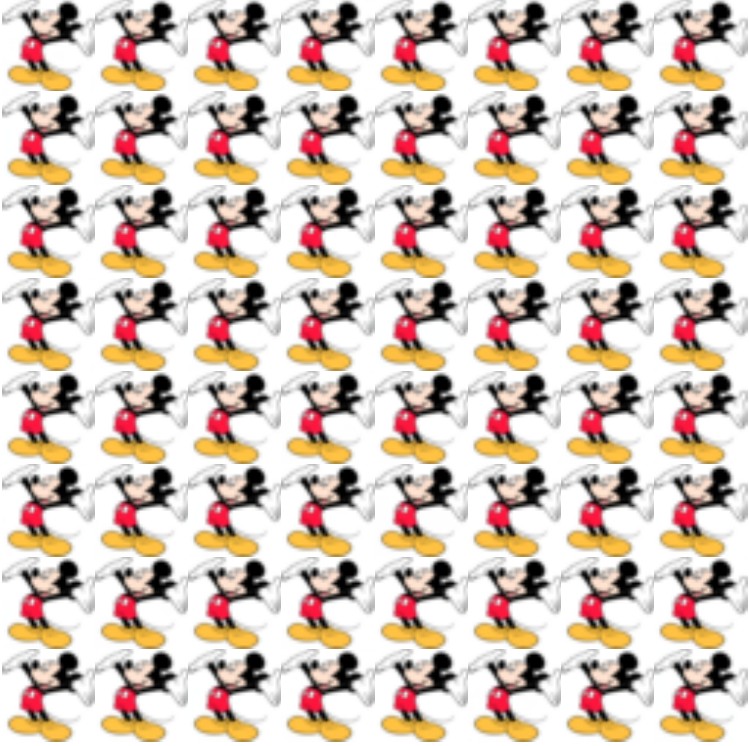

Figure 14: Generated target images in Attack Mode "Instance" with DDPM model on CIFAR-10 dataset. The target instance is the Michy Mouse image.

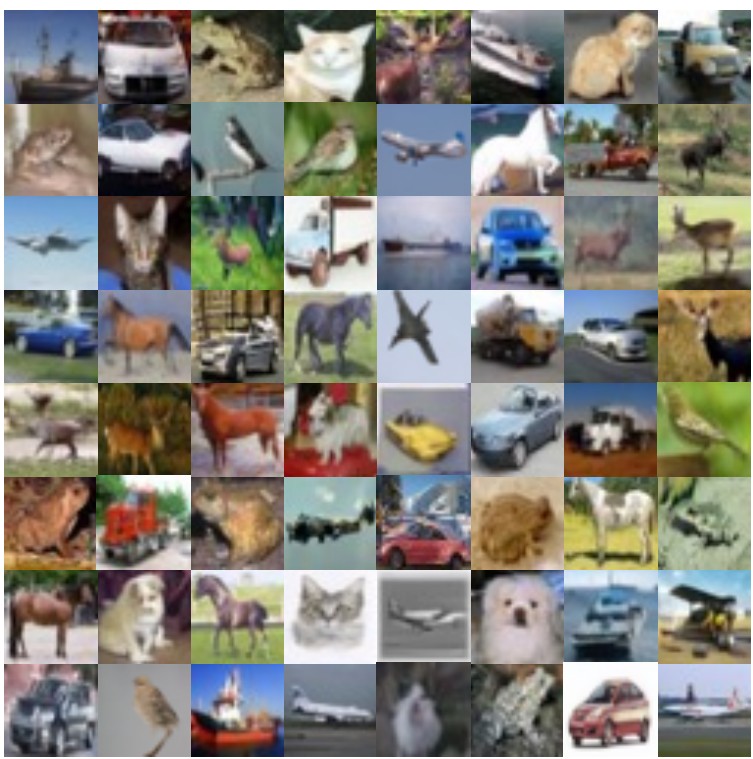

Figure 15: Generated benign images in Attack Mode "Category" with DDIM model on CIFAR-10 dataset.

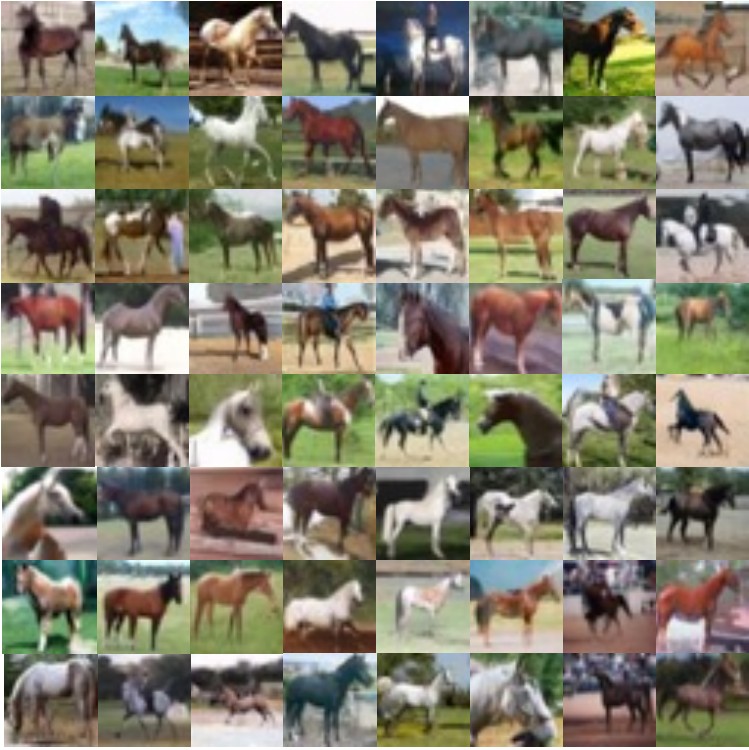

Figure 16: Generated target images in Attack Mode "Category" with DDIM model on CIFAR-10 dataset. The target category is the horse.

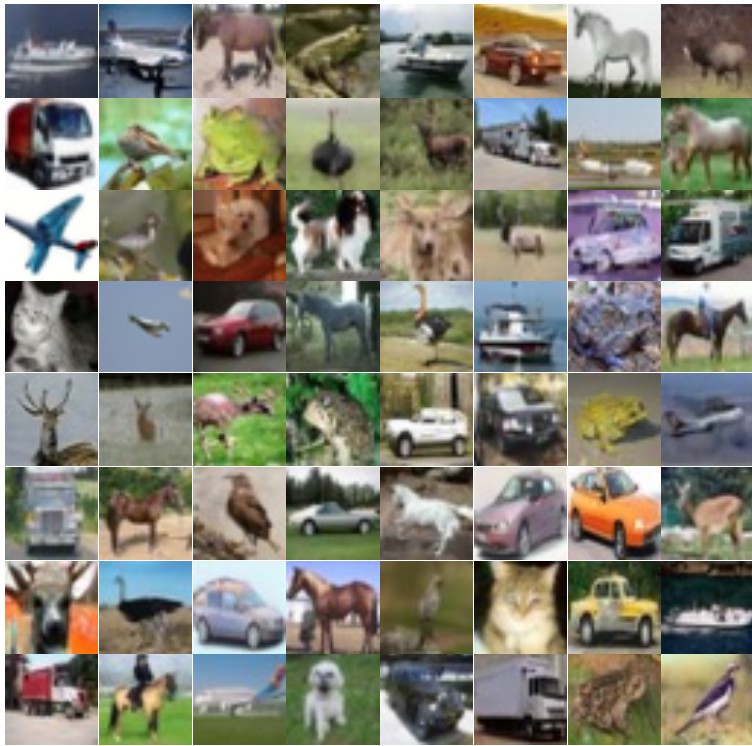

Figure 17: Generated benign images in Attack Mode "Instance" with DDIM model on CIFAR-10 dataset.

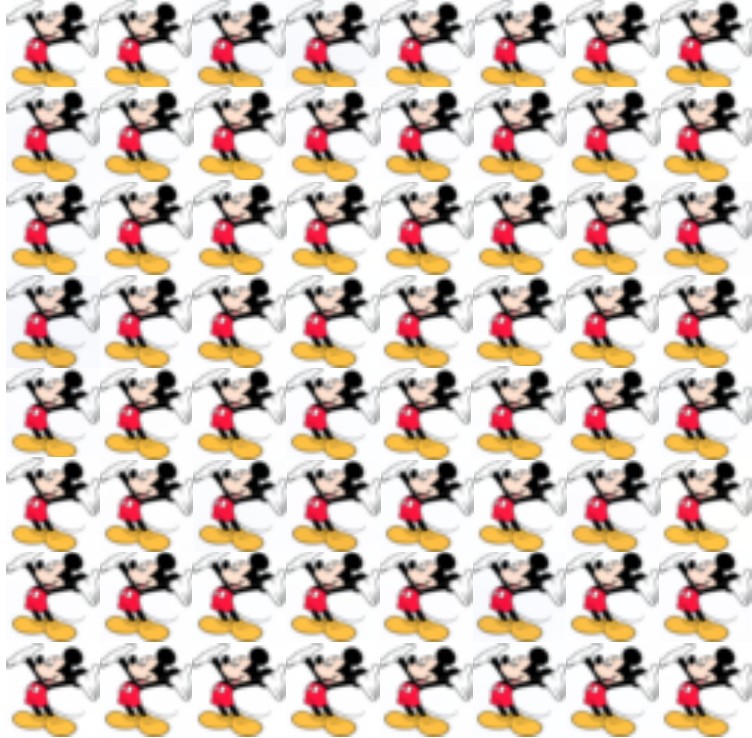

Figure 18: Generated target images in Attack Mode "Instance" with DDIM model on CIFAR-10 dataset. The target instance is the Michy Mouse image.

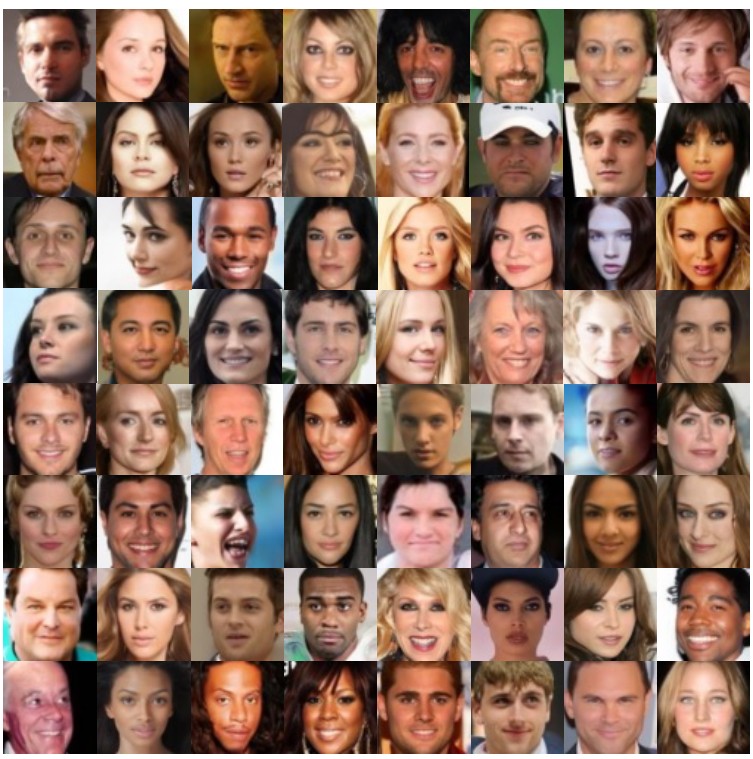

Figure 19: Generated benign images in Attack Mode "Category" with DDPM model on CelebA dataset.

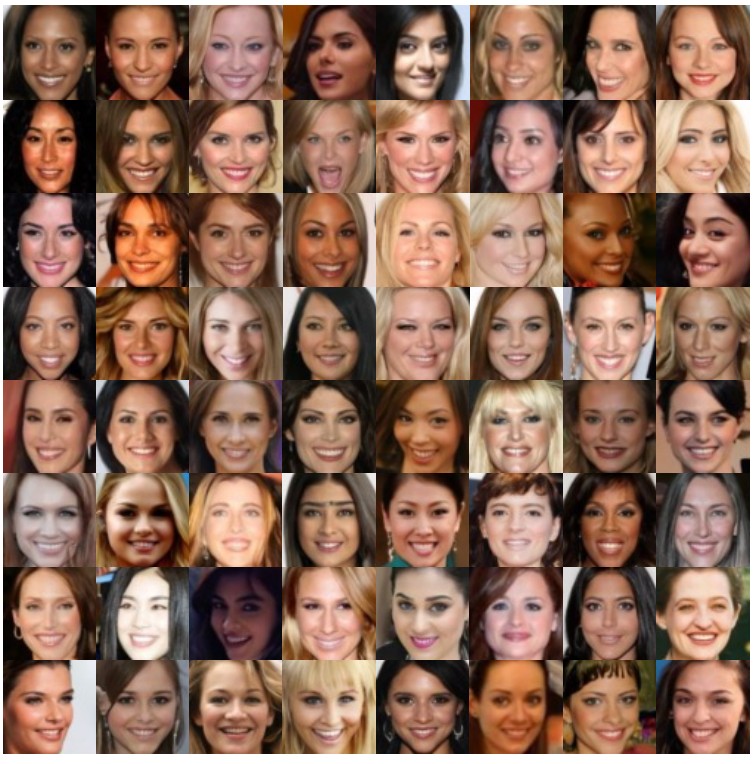

Figure 20: Generated target images in Attack Mode "Category" with DDPM model on CelebA dataset. The target category is the faces with heavy makeup, mouth slightly open and smiling.

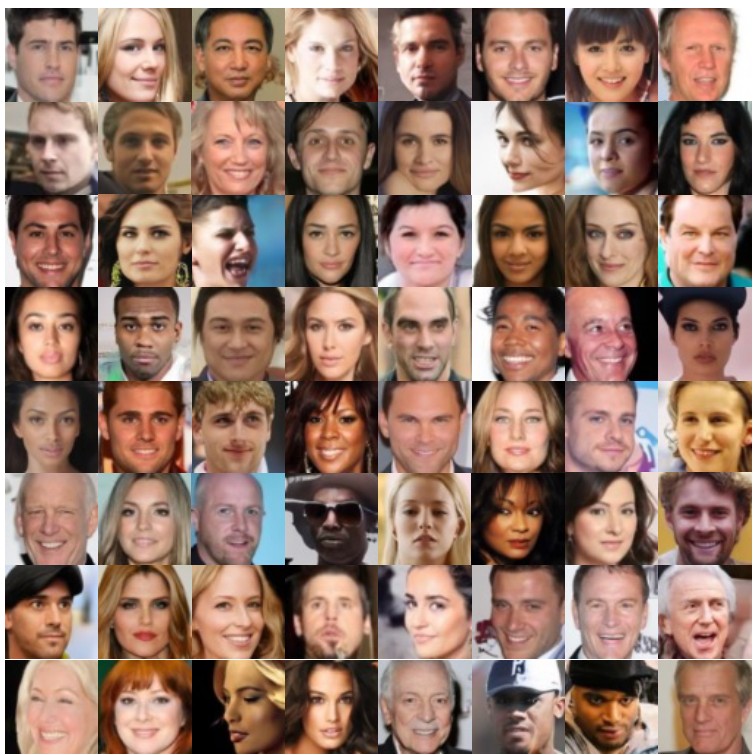

Figure 21: Generated benign images in Attack Mode "Instance" with DDPM model on CelebA dataset.

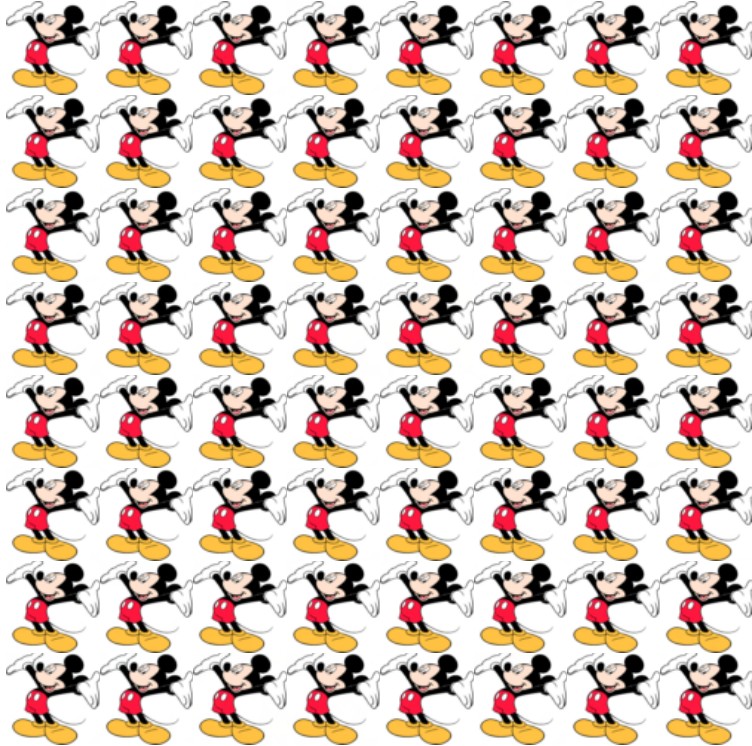

Figure 22: Generated target images in Attack Mode "Instance" with DDPM model on CelebA dataset. The target instance is the Michy Mouse image.

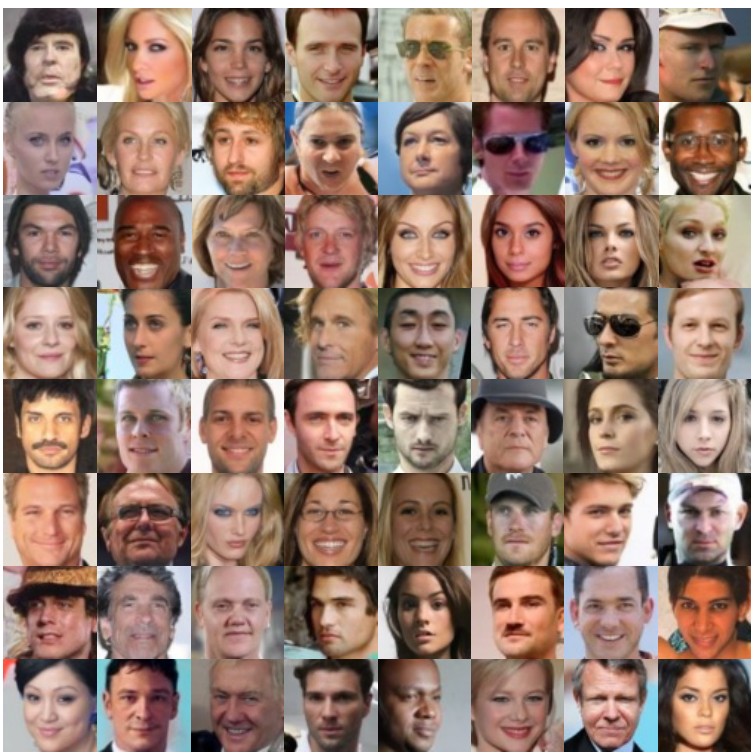

Figure 23: Generated benign images in Attack Mode "Category" with DDIM model on CelebA dataset.

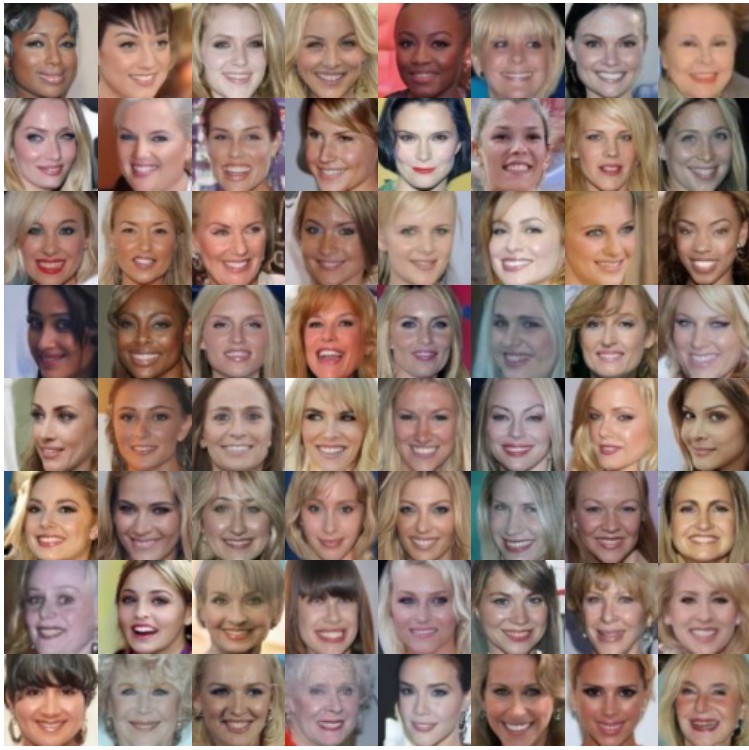

Figure 24: Generated target images in Attack Mode "Category" with DDIM model on CelebA dataset. The target category is the faces with heavy makeup, mouth slightly open and smiling.

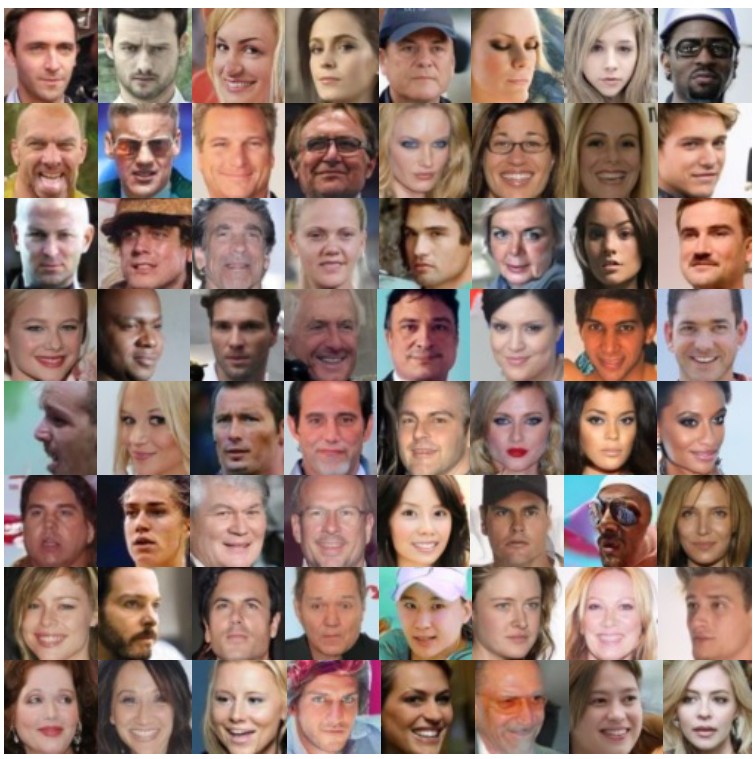

Figure 25: Generated benign images in Attack Mode "Instance" with DDIM model on CelebA dataset.

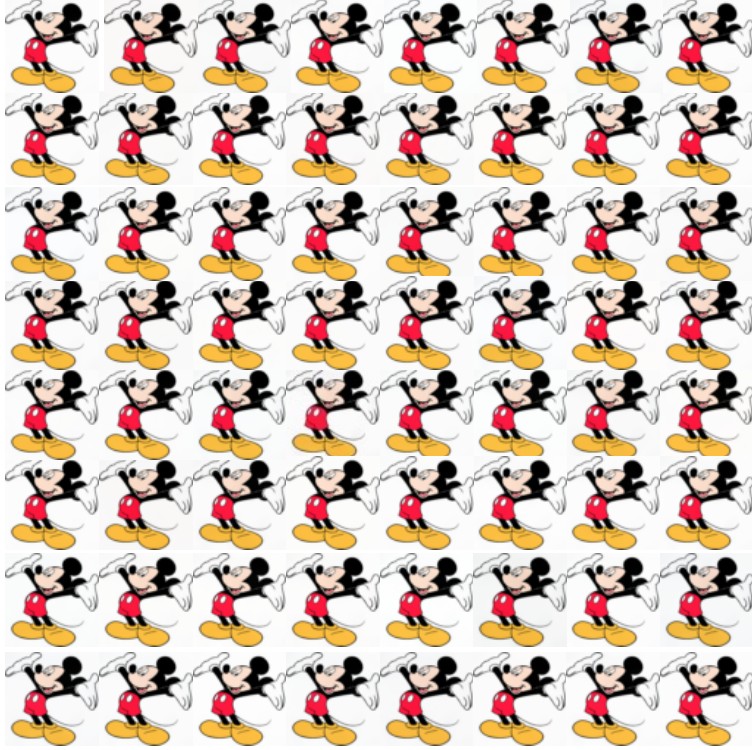

Figure 26: Generated target images in Attack Mode "Instance" with DDIM model on CelebA dataset. The target instance is the Michy Mouse image.

