# OpenReview forum: "DisDet: Exploring Detectability of Backdoor Attack on Diffusion Models"
_TMLR — Accepted by TMLR_

### Review · Reviewer_rYvA · 2024-11-27

**Summary Of Contributions:**

This paper explores backdoor attacks on diffusion models, proposing a detection method based on distributional differences to identify backdoored inputs, as well as developing a possible attack to evade this detection.

**Audience:**

Yes

**Claims And Evidence:**

Yes

**Requested Changes:**

This paper presents both a defense method and a learnable backdoor attack method. While I personally feel that the technical novelty of this work is somewhat limited, it offers a well-rounded narrative and strengthens its persuasiveness through thorough experimental evaluation. Overall, I believe this paper provides valuable insights into the field of backdoor attacks and defenses for diffusion models.

**Strengths And Weaknesses:**

### Strengths
- The paper provides a thorough analysis of the characteristics of existing fixed trigger patterns and their distribution discrepancies.
- The paper conducts extensive experiments on diffusion models and datasets, demonstrating the effectiveness of both the detection and detection-evading strategies.
- This paper is well-structured and easy to understand.

### Weaknesses
- The attack investigated in this paper is like an extension of [1][2].
- Some issues of arrangement (e.g. Table 2 extends beyond the scope).

[1] TrojDiff: Trojan Attacks on Diffusion Models with Diverse Targets

[2] How to Backdoor Diffusion Models?

---

> ### Author Response · Authors · 2025-03-11
> **Response**
>
> **Q1: Extension of [1][2].**
>
> Thank you for your comment. **Previous works focus on the attack level, however, in our paper, we propose a detection framework against the previous attack works** and further explore another attack method based on our proposed detection method. We are glad to summarize and claim our contributions in two aspects: **(I) Algorithm Level**: we propose differential distribution optimization with noise consistency optimization. **(II) Paradigm Level**: we 1) directly detect the poisoned input by analyzing the distribution shift. 2) consider the stealthiness of the trigger in the backdoored diffusion models.
>
> **Q2: Arrangement.**
>
> Thank you for pointing it out. We updated Table 2 by reducing its size in the paper.
>
> **Requested Changes:**
>
> Thank you for your thoughtful and encouraging feedback. We truly appreciate your recognition of our work. We hope our findings can contribute to the backdoor attack.

---

### Review · Reviewer_6VCm · 2024-12-08

**Summary Of Contributions:**

This paper found that the existing backdoor attacks on diffusion models often incurs a backdoored noise distribution that is different from a standard normal distribution. Therefore, this paper proposes a backdoor sample detection algorithm that removes points significantly deviated from a normal distribution. Furthermore, it also proposes a backdoor attack algorithm that can evade the previous detection by jointly minimizing the distributional difference and the diffusion model's reconstruction error.

**Audience:**

Yes

**Claims And Evidence:**

Yes

**Requested Changes:**

Please see sections above.

**Strengths And Weaknesses:**

**Strengths:**

1. The proposed methods are well-motivated and are easy to implement. The paper is also well-organized and easy to follow.

2. The experiment results seem to imply a superior performance of the proposed attack method, compared to the existing ones.

**Weaknesses:**

While I believe it is a good work, it would be great if the authors can address my following questions.

1. If the attacks such as Trojdiff decrease the magnitude of the backdoor trigger, it seems they can also bypass the proposed defense mechanism. The authors may want to present the trade-off between the ASR under the proposed detection algorithm and the backdoor model's performance for different magnitude levels.

2. Related to point 1, typically a backdoor trigger with a larger magnitude leads to a better model performance on both benign and backdoor inputs. However, Table 2 indicates the proposed attack yields a backdoor model with a better generation performance, which is not intuitive for me. Can authors kindly explain the insight behind?

3. Is there any potential defense against the proposed attack method? It will be a significant complement to this work. Otherwise, I would suggest the authors to tune down the claims such as "systematically explore the detectability of poisoned noise ..."

4. Can authors further clarify the insight behind using the NC loss? With PDD loss only, the learned trigger is not optimized to maximize backdoored model's performance. Therefore, the NC loss in the trigger learning stage need to enforce an efficient trigger.
My understanding is that this loss aims to minimize the reconstruction error between the image generated from a backdoored input and the target image (in other words, the difference between the predicted noise and a standard Gaussian noise) based on a fixed diffusion model.  It is unclear why this loss makes the trigger more efficient. The authors explanation in Section 5.2 and 7 are not sufficient to justify the use of such loss.

5. Based on point 4, what if the fixed diffusion model uses a different archtecture from the one used to train the backdoored model? As in most cases the attacker can only alter the training points but have no access to the model training process.

6. What is the needed propotion of backdoor points for a successful attack? In this paper, it seems the benign and backdoor points are 50-50.

---

> ### Author Response · Authors · 2025-03-11
> **Response**
>
> **Q1 and Q2: Better performance when adding a trigger.**
>
> Thank you for your question. Typically, a backdoor trigger with a larger magnitude cannot lead to a better performance on both benign and backdoor inputs. However, as shown in Table 2, the proposed attack yields a backdoor model with a better generation performance. This phenomenon also happens in previous works such as Trojdiff [1] and Baddifusion [2], which can achieve even better performance with a trigger. We believe that there are two potential reasons. **First**, one possible explanation is that the introduced trigger subtly modifies the model latent space in a way that enhances its ability to generate high-quality outputs. Similar effects have been observed in previous works, where small perturbations or auxiliary signals can sometimes act as implicit regularizers, improving generalization or stabilizing training dynamics. **Second**, while FID is a useful metric for evaluating the generation quality of diffusion models, it is not the most accurate, as discussed in SDXL [3] and PickScore [4]. We believe that certain variations in FID do not always provide a definitive decision on which generation quality is better. Overall, this phenomenon is intriguing and can be discussed further in the future.
>
> [1] Chen, Weixin, et al. Trojdiff: Trojan attacks on diffusion models with diverse targets. CVPR, 2023.
>
> [2] Chou, Sheng-Yen, et al. How to backdoor diffusion models? CVPR, 2023.
>
> [3] Podell, Dustin, et al. SDXL: Improving latent diffusion models for high-resolution image synthesis. ICLR 2024.
>
> [4] Kirstain, Yuval, et al. Pick-a-pic: An open dataset of user preferences for text-to-image generation. NeurIPS 2023.
>
> **Q3: Tune down the claims.**
>
> Thank you for your suggestion. Following your suggestion, we updated the paper by tuning down the claims as "In this paper, we explore the detectability of poisoned noise input for the backdoored diffusion models."
>
> **Q4: Insight behind NC loss.**
>
> We use NC loss in the trigger training stage to **ensure that the trigger adapts well to the near-pretrained diffusion model**. To maintain strong benign performance in the diffusion model, the training principle is to minimize weight updates relative to the pre-trained model. This regularized trigger effectively activates the backdoor in models that remain closely aligned with the pre-trained diffusion model. Hence, in Section 5.2, we claim "with NC loss, slight update from the original benign model may be already sufficient for fitting poisoned data samples".
>
> Without NC loss, the trigger lacks regularization and does not consider benign performance. As a result, the diffusion model would require significant modifications to achieve the attack objective, ultimately compromising its benign performance.
>
> We hope this explanation clarifies the issue and enhances understanding. We updated the paper in Section 5.2 by adding this detailed explanation to further improve its clarity.
>
> **Q5: Black-box attack.**
>
> Thank you for your thoughtful question. If the fixed diffusion model has a different architecture from the one used to train the backdoored model, the effectiveness of the attack may depend on the similarity between the two architectures. If the architectures are significantly different, the backdoor may become less effective due to differences in how features are learned and represented. Following the previous backdoor diffusion works which mainly focus on full white-box attacks, our method primarily investigates their detectability and attacks under the same settings with the access to the training stage. Investigating the transferability of cross-architecture is an interesting direction for future work, and we appreciate your suggestion to explore this aspect further.
>
> **Q6: Propotion of backdoor points.**
>
> Thank you for your question. (1) We leverage the poison ratio as **10\%** for our attack. We train the backdoor model using 90\% benign data and 10\% poisoned data. (2) From the perspective of generated images, there is no specific proportion to define the successful attack. Higher ASR with better benign performance can lead to a better backdoor diffusion attack.

---

> > ### Comment · Reviewer_6VCm · 2025-03-13
> >
> > I thank the authors for their detailed responses. While most concerns are addressed,
> >
> > 1. My question 1 is not answered in the response. I noticed a similar question is raised by Reviewer yMV9.
> >
> > 2. My questions 2 says a larger trigger in magnitude leads to a better performance in general, but the authors responded in an opposite way.

---

> > > ### Author Response · Authors · 2025-03-15
> > > **Response**
> > >
> > > **Q1: Trade-off under different magnitude levels.**
> > >
> > > Thanks for your prompt response. We totally agree that as the magnitude of the trigger in TrojDiff decreases, it becomes easier to evade distribution-based detection. As a result, this also leads to a lower ASR, highlighting a trade-off between ASR and detection rate.
> > >
> > > Here we conduct the experiments and present some results in the table. In the experiments, we applied the same PDD threshold E[D] to our proposed attack method and Trojdiff, and adjusted the default magnitude of TrojDiff to different levels. Note that a higher magnitude factor corresponds to a smaller trigger magnitude. The results illustrate the trade-offs associated with different TrojDiff magnitude settings and demonstrate that our proposed attack consistently achieves a higher ASR. We included these experiments in our updated paper. Thanks again for the valuable question.
> > >
> > > | Magnitude Factor | Pass Rate | ASR   |
> > > |------------------|-----------|-------|
> > > | Trojdiff-0.95             | 76.6%     | 36.5% |
> > > | Trojdiff-0.90             | 59.3%     | 48.6% |
> > > | Trojdiff-0.85             | 4.5%      | 3.7%  |
> > > | Trojdiff-0.80             | 0%        | 0%    |
> > > | **Ours**         | 80.0%     | 64.0% |
> > >
> > > **Q2: A larger trigger in magnitude leads to a better performance**
> > >
> > > Thanks for your feedback. We would like to mention that a larger magnitude can enhance the attack performance but degrade benign performance. This is mentioned in the Trojdiff [1], which claims that, with a larger magnitude, "Trojan noise will be
> > > similar to the trigger itself, making it harder to recover the images, since there is no random space for learning and results in the trigger-contained generated images". This suggests that as the magnitude increases, benign performance would degrade.
> > >
> > > Second, our approach aims to optimize the distribution of the poisoned noise instead of reducing its magnitude. Specifically, we adjust the distribution of the poisoned noise (i.e., the trigger combined with Gaussian noise) to approach the standard Gaussian noise rather than minimizing its magnitude. From Figure 7 in the main paper, the magnitude of our trigger increases during PDD-oriented training.
> > >
> > > Thanks again for the valuable questions.
> > >
> > > [1] Chen, Weixin, et al. Trojdiff: Trojan attacks on diffusion models with diverse targets. CVPR, 2023.

---

> ### Comment · Reviewer_6VCm · 2025-03-16
>
> I appreciate the additional experimental results, which address my question 1.
>
> I raised the second point since [1] demonstrates that a larger magnitude enhances performance on both clean and backdoored data. As authors acknowledged that "the magnitude of our trigger increases during PDD-oriented training", I feel like it aligns with the findings in [1].
>
> [1] Wang G, Xian X, Srinivasa J, et al. Demystifying poisoning backdoor attacks from a statistical perspective. ICLR 2024

---

> > ### Author Response · Authors · 2025-03-17
> > **Response**
> >
> > Dear Reviewer 6VCm,
> >
> > Thank you for your valuable feedback.
> >
> > We're glad to hear that the experimental results addressed your question.
> >
> > For the second point, we agree that our findings align with those in [1], and we appreciate your summary. We will incorporate these combined insights into the revised paper.

---

### Review · Reviewer_yMV9 · 2025-03-04

**Summary Of Contributions:**

This paper investigates the failure of backdoor attacks on diffusion models when faced with simple defenses. The authors highlight that previous attack methods primarily contaminate the input random vector of generator samples by adding noise. By comparing the distribution of these inputs with a clean Gaussian distribution, they demonstrate that discrepancies between these distributions serve as an indicator of input poisoning, allowing for detection.

To formally capture this discrepancy, the authors introduce PDD (Poisoned Distribution Discrepancy) as a measure of how much the poisoned inputs deviate from the expected benign distribution.

Furthermore, in an effort to increase the stealthiness of the trigger and enhance the effectiveness of the attack, they attempt to minimize this discrepancy while learning the optimal trigger.

**Audience:**

Yes

**Claims And Evidence:**

Yes

**Requested Changes:**

Clarify the distinguishing factors: The paper should further explore and explicitly define the characteristics that separate adversarial noise from natural noise. Are there specific statistical properties unique to poisoned noise?

Evaluate alternative defense mechanisms: Instead of solely focusing on minimizing PDD, the study could test whether conventional denoising techniques (such as Gaussian smoothing or diffusion-based purification) yield comparable results in mitigating backdoor attacks.

Analyze the trade-off: The authors should discuss potential trade-offs between attack stealthiness and detection robustness—does reducing the PDD significantly weaken the attack itself, making it easier to defend against?

**Strengths And Weaknesses:**

Strengths:

The main claim of the paper—that backdoor attack detection in diffusion models is feasible—is valid.

The proposed defense mechanism performs well, as evidenced by the provided results.


Weaknesses

One of the key challenges in this study is that not all noise can be considered a backdoor attack. The fundamental question remains:
How can we reliably distinguish between benign noise and noise that has been injected with malicious intent?


If a detection method simply identifies noise differences, it may also mistakenly flag natural variations in input randomness as an attack, leading to false positives. Therefore, it is crucial to develop a more robust approach that explicitly differentiates between harmless noise and adversarial perturbations.


Another point to consider is whether the proposed attack minimization strategy is necessary.
Could simple denoising or smoothing of the input random vector achieve similar results?
If denoising can remove the contamination effectively, then instead of minimizing PDD, a more practical defense could involve filtering or reconstructing the input to neutralize potential backdoor effects.

---

> ### Author Response · Authors · 2025-03-11
> **Response**
>
> **Q1: A more robust approach to differentiate between benign and harmful noise.**
>
> A1: Thank you for your feedback. To distinguish between benign Gaussian noise and Gaussian noise with malicious trigger noise, we show the statistical distribution difference in Figure 2. Based on this observation, we would like to emphasize that our detection method can effectively distinguish between benign and malicious noise. As demonstrated in Table 1 of the main paper, by setting the detection threshold to $E[D(x)] + 3\sigma$ (corresponding to the 3-sigma percentile, i.e., 99.9\%) in Eq. (4), our method successfully detects 100\% of malicious noise while **affecting only approximately 0.1\% of benign noise.**
>
> We updated Table 1 in the revised paper to claim this point. We appreciate your suggestion for improving the paper.
>
> **Q2: Effect of traditional denoising methods.**
>
> A2: Thank you for your suggestion. Although traditional denoising operations such as denoising, smoothing, filtering, and reconstruction may mitigate attacks, these methods alter benign noise such that it no longer follows a Gaussian distribution. Consequently, diffusion models relying on Gaussian noise fail to generate realistic images and instead produce random, non-realistic outputs, severely compromising the generative process. Therefore, our proposed PDD detection remains a practical and effective technique for detecting malicious noise.
>
> We updated the paper by adding the experiments shown in "Other Techniques for Removing Backdoored Triggers." and Table 6. As a result, **using traditional denoising operations severely degrades the quality of the generated benign images.**
>
> On the other hand, the goal of **"attack minimization" is to help us design and propose an attack method** to bypass the proposed distribution-based detection system, which is not a detection method. To clarify, our paper first introduces a distribution-based detection method and subsequently presents an attack method capable of overcoming this detection approach. We provide this integrated perspective to the security community with insights for the better design of detection frameworks in the future.
>
> **Requested Changes:**
>
> **1. Clarify the poisoned noise and natural noise.**
>
> Thank you for your feedback. We updated Table 1 of the main paper to highlight this point according to the above Q1. To distinguish between benign Gaussian noise and Gaussian noise with malicious trigger noise, we show the statistical distribution difference in Figure 2. Based on this observation, we would like to emphasize that our detection method effectively distinguishes between benign and malicious noise. By setting the detection threshold to $E[D(x)] + 3\sigma$ (corresponding to the 3-sigma percentile, i.e., 99.9\%) in Eq. (4), our method successfully detects 100\% of malicious noise while \textbf{affecting only approximately 0.1\% of benign noise}. We appreciate your suggestion for improving the paper.
>
> **2. Evaluate alternative defense mechanisms.**
>
> Thank you for your suggestion. According to your suggestion, we updated our paper to compare the conventional denoising techniques and ours in Table 6 in the main paper according to the above Q2. Although conventional denoising operations may mitigate attacks, these methods alter benign noise such that it no longer follows a Gaussian distribution. Consequently, diffusion models relying on Gaussian noise fail to generate realistic images and instead produce random, non-realistic outputs, severely compromising the generative process. Therefore, our proposed PDD detection remains a practical and effective technique for detecting malicious noise.
>
> **3. Analyze the trade-off.**
>
> Thank you for your advice. According to your suggestion, we updated the paper by adding the "Discussion of the trade-off between attack capability and stealthiness" in Section 7. A higher level of attack stealthiness makes the poisoned noise more closely to standard Gaussian noise, which makes it harder to detect or defend against. However, this also weakens the effectiveness of the trigger, causing more benign images to be generated and reducing the overall impact of the attack. Conversely, lowering stealthiness increases the strength of the attack but makes the noise more obvious and easier to detect.

---

> > ### Comment · Reviewer_yMV9 · 2025-04-03
> >
> > I agree with the authors that denoising, smoothing, filtering may unintentionally distort the natural statistics of benign inputs — particularly when the noise follows a Gaussian distribution. That said, it is indeed possible to preserve such statistical properties through careful design choices. For instance, one could incorporate statistical regularization during the training of the denoiser to maintain Gaussian characteristics, utilize probabilistic denoising methods (e.g., Bayesian or diffusion-based models), or apply lightweight and adaptive filtering to minimize unnecessary alteration of benign noise. Alternatively, learning noise-invariant representations can reduce the reliance on explicit preprocessing. Post-denoising statistical correction is another viable option to restore the original noise distribution. I encourage the authors to evaluate whether their defense mechanism preserves the natural distribution of benign noise — particularly Gaussianity — and to consider incorporating one or more of these strategies to avoid unintended distributional shifts

---

> > > ### Author Response · Authors · 2025-04-03
> > >
> > > Thank you for your summary and thoughtful response. Please allow us to briefly recap the comments.
> > >
> > > Firstly, the reviewer expressed curiosity regarding whether simple denoising or smoothing of the input random vector could achieve comparable results to show the necessity of our proposed attack minimization strategy. **To address this, we included Table 6 in our manuscript, demonstrating that traditional denoising methods significantly degrade the quality of the generated benign images. This verifies the necessity of our proposed attack minimization strategy. We note that the reviewer agrees with this observation.**
> > >
> > > Based on this conclusion, the reviewer further suggested exploring whether more sophisticated, carefully designed filtering or denoising methods could preserve the quality of the generated benign images.
> > >
> > > While we appreciate this suggestion, **we respectfully emphasize: First, our proposed defense method already effectively detects nearly all previously known attacks, such as Trojdiff, while preserving the natural distribution of benign noise through a straightforward and efficient design. Second, our proposed defense method already effectively satisfies the suggestion "avoid unintended distributional shifts".** Although more complicated denoising or filtering methods might also achieve the same effect, they remain underexplored in this scenario. Therefore,  these more complex methods could be conducted as a separate research work in the future compared to our current work. We appreciate your valuable suggestions. Please let me know if you have further questions.

---

### Decision · Action_Editor_cg4d · 2025-05-07

**Recommendation:** Accept with minor revision

**Comment:**

This paper investigates the failure of backdoor attacks on diffusion models when facing simple defenses. It shows that existing attacks often introduce a backdoored noise distribution that deviates from the standard normal distribution. Based on this observation, the authors propose a detection method that filters out inputs significantly deviated from a clean Gaussian. They further develop a new attack algorithm that jointly minimizes the distributional difference and the diffusion model’s reconstruction error, allowing it to avoid the proposed detection. This work highlights both the weaknesses of existing attacks and the potential for stronger defenses.

Overall, the reviewers find the work interesting and acknowledge its contributions. Moreover, most of the reviewers' comments have been addressed during the rebuttal. One reviewer still raises concerns regarding the comparison of the proposed defense method with more sophisticated denoising approaches. In the revision, the proposed defense demonstrates its effectiveness using simple filtering methods. However, I also agree that it would be beneficial to mention potential directions for more advanced filtering techniques in the conclusion or appendix. Overall, the paper meets the acceptance criteria of TMLR, and I recommend its acceptance after the authors include future work in the conclusion.

**Audience:**

Backdoor attacks are a popular research topic, and the TMLR audience is likely to find this work interesting.

**Claims And Evidence:**

The claims are well supported by the experiments.

---

> ### Author Response · Authors · 2025-06-06
> **Camera-ready Revision.**
>
> We thank AE and all reviewers for valuable feedback. In response to the review, we have updated the paper to include potential directions for more advanced filtering techniques in the conclusion.